# The transcriptional landscape underlying larval development and metamorphosis in the Malabar grouper (*Epinephelus malabaricus*)

Roger Huerlimann[1,2]*[†], Natacha Roux[3][†], Ken Maeda[4], Polina Pilieva[4], Saori Miura[4], Hsiao-chian Chen[1,4], Michael Izumiyama[1], Vincent Laudet[4,5][‡], Timothy Ravasi[1,6][‡]

[1]Marine Climate Change Unit, Okinawa Institute of Science and Technology Graduate University, Onna-son, Japan; [2]Centre for Sustainable Tropical Fisheries and Aquaculture, College of Science and Engineering, James Cook University, Townsville, Australia; [3]Computational Neuroethology Unit, Okinawa Institute of Science and Technology Graduate University, Onna-son, Japan; [4]Marine Eco-Evo-Devo Unit, Okinawa Institute of Science and Technology Graduate University, Onna-son, Japan; [5]Marine Research Station, Institute of Cellular and Organismic Biology, Academia Sinica, Jiau Shi, Taiwan; [6]Australian Research Council Centre of Excellence for Coral Reef Studies, James Cook University, Townsville, Australia

*For correspondence:
roger.huerlimann@oist.jp

[†]These authors contributed equally to this work

[‡]Equal last authors

**Competing interest:** The authors declare that no competing interests exist.

**Abstract** Most teleost fishes exhibit a biphasic life history with a larval oceanic phase that is transformed into morphologically and physiologically different demersal, benthic, or pelagic juveniles. This process of transformation is characterized by a myriad of hormone-induced changes, during the often abrupt transition between larval and juvenile phases called metamorphosis. Thyroid hormones (TH) are known to be instrumental in triggering and coordinating this transformation but other hormonal systems such as corticoids, might be also involved as it is the case in amphibians. In order to investigate the potential involvement of these two hormonal pathways in marine fish post-embryonic development, we used the Malabar grouper (*Epinephelus malabaricus*) as a model system. We assembled a chromosome-scale genome sequence and conducted a transcriptomic analysis of nine larval developmental stages. We studied the expression patterns of genes involved in TH and corticoid pathways, as well as four biological processes known to be regulated by TH in other teleost species: ossification, pigmentation, visual perception, and metabolism. Surprisingly, we observed an activation of many of the same pathways involved in metamorphosis also at an early stage of the larval development, suggesting an additional implication of these pathways in the formation of early larval features. Overall, our data brings new evidence to the controversial interplay between corticoids and thyroid hormones during metamorphosis as well as, surprisingly, during the early larval development. Further experiments will be needed to investigate the precise role of both pathways during these two distinct periods and whether an early activation of both corticoid and TH pathways occurs in other teleost species.

## eLife assessment

The work provides **valuable** genomic resources to address the endocrine control of a life cycle transition in the Malabar grouper fish. The revised manuscript is more **solid** and the resources and experimental data help to build up a meaningful biological understanding of thyroid signaling in grouper fish.

## Introduction

Most teleost fishes have a stage-structured life cycle that includes a transition between larval and juvenile phases known as metamorphosis; this transition is regulated by TH (*Laudet, 2011*; *McMenamin and Parichy, 2013*). Of all teleost fishes, flatfishes experience one of the most extreme metamorphosis, with significant changes occurring in their body organization and appearance during this period, switching from a symmetrical to an asymmetrical body plan (*Schreiber, 2013*; *Shao et al., 2017*). However, metamorphic changes are not always as pronounced in other fish species. For example, the metamorphosis of zebrafish is mainly marked by relatively discrete pigmentation changes that appear to be regulated by TH (*Brown, 1997*; *Guillot et al., 2016*; *McMenamin et al., 2014*; *Walpita et al., 2009*).

Metamorphosis in teleost fishes is not only marked by visible changes in the body, but also by a range of ecological, physiological, biochemical, and behavioral changes. These changes are thought to be initiated and coordinated by a surge of TH, which regulates various signaling pathways through the action of specific transcription factors known as thyroid hormone receptors (TRα, TRβ). For example, there is evidence that TH is associated with the transition between oceanic and coral reef environments in the convict surgeonfish (*Holzer et al., 2017*), controls pigmentation changes in zebrafish, clownfish, and grouper (*Salis et al., 2021*; *Saunders et al., 2019*), regulates ossification processes in zebrafish and flatfishes (*Campinho et al., 2018*; *Pelayo et al., 2012*) and is involved in the shift of visual perception by controlling the expression of opsin genes in many species (*Roux et al., 2023*; *Volkov et al., 2020*). More recently, it has also been suggested that the metabolic changes that occur during larval development in teleosts may be regulated by TH, as demonstrated in clownfish (*Roux et al., 2023*). Of note in some cases, like groupers (elongated spines) (*Colin and Koenig, 1996*; *Kohno et al., 1993*; *Powell and Tucker, 1992*) or carapids (vexillum appendage) (*Govoni, 1984*), some changes occur very early on and are considered as temporary specialization of the pelagic larval stages, serving as anti-predator defense (*Moster, 1981*), flotation (*Nonaka et al., 2021*) or camouflage (*Leis and Carson-Ewart, 2000*). It is still unclear if these are late developmental processes occurring after hatching or early manifestations of metamorphosis.

Besides the TH signaling pathway, other actors have been shown to be important in metamorphosis regulation. For example, studies have provided clear evidence that corticoids and TH are interacting together to regulate amphibian metamorphosis (*Denver, 2009*; *Paul et al., 2022*; *Sachs and Buchholz, 2019*). But as far as we know, there is limited information available regarding the interaction between corticoids and TH during fish metamorphosis. Although a synergistic effect of cortisol and TH has been observed in flatfish metamorphosis (advancement of morphological changes), there has been insufficient investigation into the communication between corticoids and TH pathways during teleost metamorphosis (*de Jesus et al., 1991*; *de Jesus et al., 1990*). More research is needed, and the use of genomic analysis would be a good way to investigate which pathways are associated with early larval development and metamorphosis.

The use of high-throughput sequencing techniques, such as transcriptomics, has made it possible to study gene expression in greater detail, especially when combined with a high-quality annotated genome, which has enabled the identification of genes that may be involved in the key biological changes that occur during metamorphosis. These techniques have provided valuable insights into the underlying molecular mechanisms that drive metamorphosis in teleost fishes (*Mazurais, 2011*). Most of the studies investigating the transcriptomic changes during marine fish larval development have been focused on commercial fish species used in aquaculture to: (*i*) gain insight into the key biological processes that occur, (*ii*) identify the genes involved in these processes, and (*iii*) find ways to improve rearing conditions to ensure high survival rates and harmonious development (*Mazurais, 2011*). However, these studies rarely mention metamorphosis to explain the onset of the various processes occurring during the transition between larval and juvenile stages. This is another reason why studying the molecular changes occurring during the larval development of the Malabar grouper *Epinephelus malabaricus* is very relevant. In addition, as mentioned above, groupers display elongate appendages during early larval development that disappear over time, providing an interesting way to study the development of these enigmatic structures (*Colin and Koenig, 1996*; *de Jesus et al., 1998*).

Our study will thus allow for a better understanding of the biological processes at play during the early larval development and metamorphosis, and to understand the carry-over effect in the context of aquaculture. Indeed, it is well known that rearing conditions may impact welfare and growth at later

stages and understanding the molecular changes occurring during the development of this species might be useful to enhance survival rates (*Dingeldein and White, 2016*; *Gagliano et al., 2007*; *Ward and Slaney, 1988*).

Grouper (Family Serranidae, Subfamily Epinephelinae) are a group of fish of both economic and ecological importance. Inhabiting temperate and tropical waters of eastern and southern regions Indo-Pacific region, East Atlantic, Mediterranean regions, and the intertropical American zone, they comprise 165 species in 16 genera (*Craig and Heemstra, 2011*; *Pierre et al., 2008*). Ecologically, groupers provide a wide variety of important functions as large top-level predators (*Ribeiro et al., 2021*). However, due to their high economic value on the food market, more than 40 species are at risk of extinction (*Luiz et al., 2016*; *Sadovy de Mitcheson et al., 2013*). This has led to the wide-spread development of grouper aquaculture farms, which produced 155,000 tons per year according to the Food and Agriculture Organization of the United Nations in 2015, with 95% of global production occurring in Asia (*FishStatJ, 2017*; *Rimmer and Glamuzina, 2019*). Despite wide variations in growth rate, body size, and color, groupers share many biological traits and lifestyles, such as protogynous hermaphroditism, complex social structure (*Heemstra, 1993*), and a biphasic lifestyle. Like many marine fishes, groupers larvae hatched after 24–48 hr of embryonic development giving rise to a transparent elongated larvae surrounded by an embryonic fin fold. After a couple of days melanophores colonize the tail (after the anus) and the gut. Shortly after the elongated appendages composed of two pelvic spines and the second dorsal spines both displaying melanophores at their tips appear. The elongation of these spines occurs before notochord flexion and their regression is concomitant with the appearance of the adult-like body pattern (*Kohno et al., 1993*; *Powell and Tucker, 1992*; *Hussain and Higuchi, 1980*; *Sawada et al., 1999*; *Kawabe and Kohno, 2009*). Their regression as well as the development of the adult-like body pattern has been demonstrated to be under the control of TH in *E. coioides* suggesting that it corresponds to the TH-regulated metamorphosis (*de Jesus et al., 1998*).

In order to gain insight into the molecular pathways involved in grouper larval development, we assembled a chromosome-scale genome sequence of *E. malabaricus* and conducted a transcriptomic analysis of nine developmental stages ranging from freshly hatched larvae to roughly two-month-old juveniles. We investigated the expression patterns of genes involved in the TH pathway and four biological processes known to be regulated by TH in other teleost species during the metamorphosis step: ossification, pigmentation, visual perception, and metabolic transition. In addition, we used the TH pathway and downstream-regulated biological processes activation as indicators to look for the potential involvement of corticoids during larval development. We observed the activation of the TH pathway during the regression of fin spines, which in other grouper species coincides with the surge of TH and marks the beginning of metamorphosis. Interestingly, the activation of the TH pathway at this stage was associated with the activation of corticoid pathways as well as the four biological processes we investigated. Especially noteworthy is the observation of an early activation of the two regulatory pathways (TH and corticoids) occurring before the formation of the elongated fin spines during early larval development.

## Results and discussion
### Genome assembly, phasing, scaffolding, and annotation

A total of 46 Gbp of PacBio HiFi reads (~43 X coverage, *Table 1*) were assembled into a fully haplotype phased genome of the Malabar grouper (*Epinephelus malabaricus*) with the primary phase consisting of 298 contigs across 1.09 Gbp genome length, a contig N50 of 7.4 Mbp, and a genome level BUSCO completeness of 93.6% with 1.3% duplication (*Table 2*). The raw assembly was further scaffolded by

**Table 1.** PacBio HiFi data generated for *E. malabaricus* genome assembly based on three SMRT cells.

|  | SMRT cell 1 | SMRT cell 2 | SMRT cell 3 | Total |
|---|---|---|---|---|
| ≥Q20 Reads | 322,103 | 442,205 | 1,373,662 | 2,137,970 |
| ≥Q20 Yield (bp) | 8,468,697,810 | 11,690,872,687 | 26,355,056,210 | 46,514,626,707 |
| ≥Q20 Read Length (mean, bp) | 26,291 | 26,437 | 19,185 | - |

**Table 2.** Statistics of the *Epinephelus malabaricus* chromosome-scale genome assembly, scaffolding and gene annotation.

| Contig assembly size | 1,092,599,927 bp |
|---|---|
| Number of contigs | 298 |
| Contig N50 | 7,396,124 bp |
| Largest contig | 26,202,351 bp |
| Mean base-level coverage PacBio HiFi | 43 X |
| Contig length contained in scaffolds | 92.8% |
| Contigs contained in scaffolds | 90.5% |
| Scaffolded assembly size | 1,027,628,325 bp |
| Number of scaffolds | 24 |
| Scaffold N50 | 43,313,630 bp |
| Largest Scaffold | 50,623,973 bp |
| Smallest Scaffold | 22,540,365 bp |
| Non-ATGC characters | 36,700 bp (0.003%) |
| GC contents | 41.3% |
| Genome: BUSCO completeness | 3,406 (93.6%) |
| Genome: Complete and single copy | 3,359 (92.3%) |
| Genome: Complete and duplicated | 47 (1.3%) |
| Genome: Fragmented | 48 (1.3%) |
| Genome: Missing | 186 (5.1%) |
| Number of protein-coding genes | 26,140 |
| Average gene length | 20,718 bp |
| Average CDS length | 1,750 bp |
| Average exons per gene | 11.2 |
| Repeat contents (DFAM) | 56.4 % |
| Number of protein-coding genes | 26,140 |
| Gene annotation: BUSCO completeness | 3476 (95.5%) |
| Gene annotation: Complete and single copy | 3,429 (94.2%) |
| Gene annotation: Complete and duplicated | 47 (1.3%) |
| Gene annotation: Fragmented | 31 (0.9%) |
| Gene annotation: Missing | 133 (3.6%) |

Phase Genomics using HiC data, resulting in a 1.03 Gbp assembly across 24 pseudo-chromosomes (*Table 2*). The scaffolded pseudo-chromosomes ranged from 22.5 Mbp to 50.6 Mbp in size and contained 90.5% of the contigs and 92.8% of the contig length (*Figure 1*). The gene model annotation resulted in 26,140 protein-coding genes, with a BUSCO completeness of 95.5% and a duplication level of 1.3%. The final GC content was 41.3% and the assembly contained 56.4% repeat regions overall, which were mainly made up of DNA transposons (28.9%), followed by LINEs (5.3%), and LTR elements (2.2%) (*Table 3*). The genome length, GC content, repeat content, number of gene models, and BUSCO values are similar to other published chromosome-level grouper genomes, for example *Epinephelus lanceolatus* (*Zhou et al., 2019*), *E. akaara* (*Ge et al., 2019*), and *E. moara* (*Zhou et al., 2021*).

## General transcriptomic results

Transcriptomic analysis of *E. malabaricus* larval development was performed on grouper larvae raised in the Okinawa Prefectural Sea Farming Center. An average of 77.1 M reads were obtained per

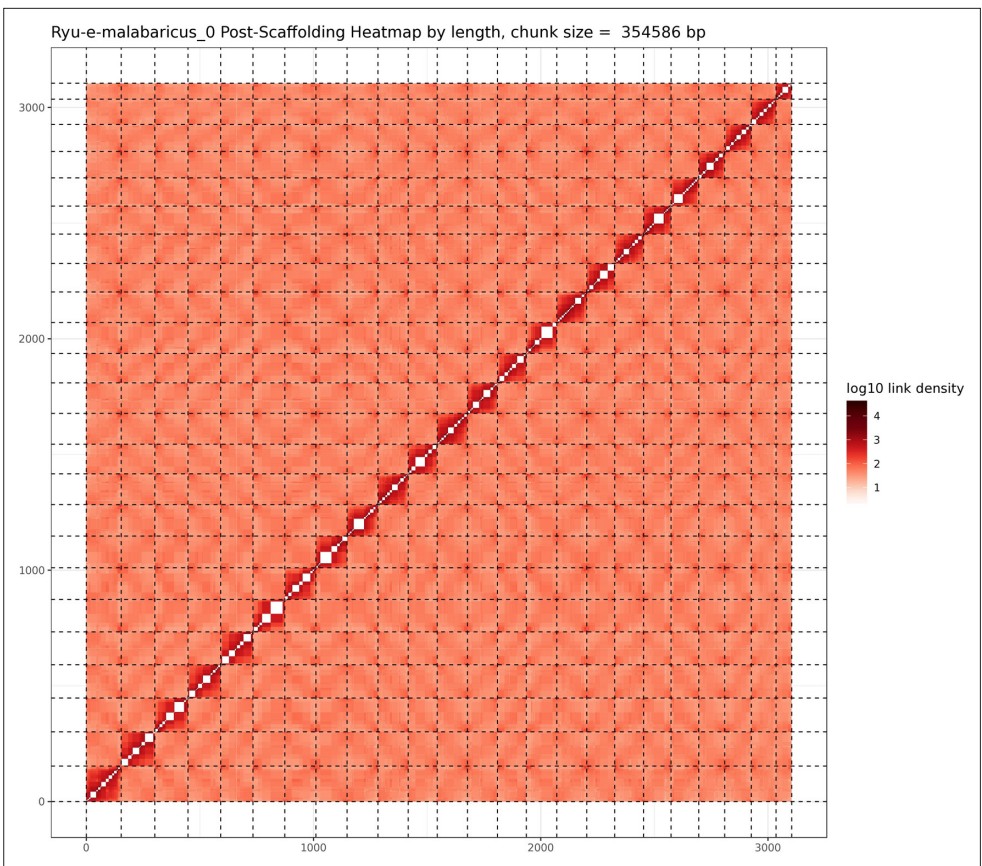

**Figure 1.** Hi-C contact map after scaffolding. *E. malabaricus* genome contig contact matrix using Hi-C data. The color bar indicates contact density from dark red (high) to white (low).

sample (pooled or individual entire larvae), which after quality control and mapping resulted in an average of 65.8 M uniquely mapped reads (85.6%) per sample for differential gene analysis. Sampled larvae from one day to two months old were sorted according to their morphology allowing us to sequence nine developmental stages (D01, D03, D06, D10, D13, D18, D32, D60, Juvenile) (*Table 2*). Principal component analysis (PCA) performed on all genes allowed to distinguish between three distinct groups: early developmental phase (composed of D01), intermediate developmental phase (composed of D03, D06, D10, D13, and D18) and late developmental phase (composed of D32, D60, and Juvenile) (*Figure 2A*).

The analysis of upregulated genes during this post-embryonic development series revealed two major peaks of gene expression that underlie the clusters of regulated genes. Indeed, the cluster analysis shows 2651 genes upregulated on D03 and to a lesser degree on D32 (clusters 1 and 2), 1515 genes upregulated on D32 (cluster 3), and 785 genes upregulated on D32 and to a lesser degree on D03 (cluster 4) (*Figure 2B*). Unsurprisingly, these two transitions, D01 to D03 and D18 to D32, also show the highest number of differentially expressed genes with 14,830 genes (7151 up, 7,679 down) between D01 and D03, and 10,774 genes (5320 up and 5454 down) between D18 and D32 (*Supplementary file 1*). This suggests that there are two major events occurring in terms of gene expression: one early on, at day 3, and one later around day 32. This last event corresponds to the separation between the intermediate and late developmental phases and is concomitant with the regression of the elongated spines, an overall change of shape, and progression of the pigmentation. In other grouper species, the regression of the elongated spines corresponds to the onset of metamorphosis and is associated with an increase in TH levels (*de Jesus et al., 1998*). However, the very early event is more striking as such a global gene expression change very early on has, to our knowledge, never been reported in other teleost fish species.

**Table 3.** Detailed repeat annotation results using the DFAM repeat database.

| Total Genome length | 1,027,628,325 bp | | |
| --- | --- | --- | --- |
| Bases masked | 579,515,295 bp (56.4 %) | | |
| | number of elements | length occupied | percentage of sequence |
| Retroelements | 660,880 | 123,046,856 | 11.97% |
| SINEs: | 74,049 | 7,798,000 | 0.76% |
| Penelope | 25,476 | 3,705,071 | 0.36% |
| LINEs: | 422,597 | 86,199,668 | 8.39% |
| CRE/SLACS | 1 | 100 | 0.00% |
| L2/CR1/Rex | 266,320 | 53,565,106 | 5.21% |
| R1/LOA/Jockey | 10,918 | 2,170,694 | 0.21% |
| R2/R4/NeSL | 11,286 | 3,660,204 | 0.36% |
| RTE/Bov-B | 47,451 | 10,563,888 | 1.03% |
| L1/CIN4 | 35,250 | 8,411,019 | 0.82% |
| LTR elements: | 164,234 | 29,049,188 | 2.83% |
| BEL/Pao | 10,186 | 2,183,697 | 0.21% |
| Ty1/Copia | 4,658 | 823,228 | 0.08% |
| Gypsy/DIRS1 | 76,319 | 13,796,105 | 1.34% |
| Retroviral | 32,317 | 5,441,530 | 0.53% |
| | | | |
| DNA transposons | 1,604,009 | 296,558,272 | 28.86% |
| hobo-Activator | 799,713 | 138,795,609 | 13.51% |
| Tc1-IS630-Pogo | 138,975 | 24,805,298 | 2.41% |
| PiggyBac | 22,605 | 3,909,912 | 0.38% |
| Tourist/Harbinger | 161,216 | 38,134,353 | 3.71% |
| Other (Mirage, P-element, Transib) | 52,802 | 10,665,183 | 1.04% |
| | | | |
| Rolling-circles | 101,574 | 30,491,612 | 2.97% |
| | | | |
| Unclassified: | 669,818 | 111,877,701 | 10.89% |
| | | | |
| Total interspersed repeats: | | 531,482,829 | 51.72% |
| | | | |
| Small RNA: | 26,405 | 2,780,064 | 0.27% |
| | | | |
| Satellites: | 11,580 | 2,523,082 | 0.25% |
| Simple repeats: | 277,904 | 12,099,447 | 1.18% |
| Low complexity: | 29,110 | 1,563,810 | 0.15% |

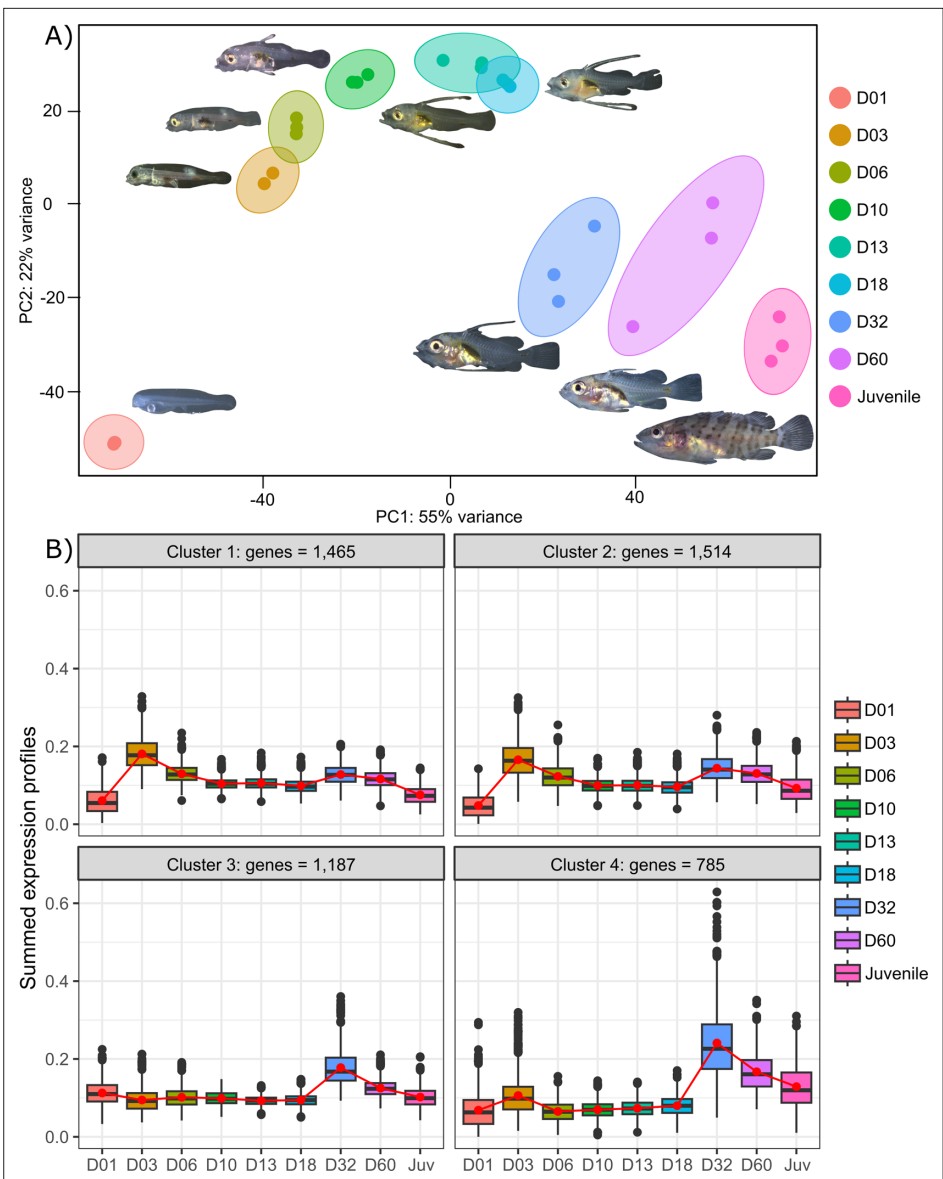

**Figure 2.** Transcriptomic results of *E. malabaricus* larval development. (**A**) Principal component analysis of different larval stages using variance stabilizing transformed complete transcriptome. (**B**) Cluster analysis using the coseq R package, focusing on genes that are upregulated on days 3 and/or 32. The number of genes in each cluster are shown above each graph. Adjusted p-values and functional annotations for the four gene clusters in this figure can be found in ource data 2. Gene expression data was generated from whole larvae.

The online version of this article includes the following figure supplement(s) for figure 2:

**Figure supplement 1.** Complete cluster analysis of differentially expressed genes.

## Two periods of activation of the TH signaling pathway during grouper post-embryonic development

We investigated the expression patterns of key genes involved in the hypothalamo-pituitary-thyroid axis (*tshb, trhr1a, trhr1a-like, trhr1b, trhr2*) as well as in TH synthesis (*tg, tpo, nis*), TH metabolism (*dio1, dio2, dio3*), and finally the genes encoding thyroid hormone receptors (*trα, trαβ, trβ*). These genes all play important roles in the regulation of TH levels and TH signaling in the body and understanding their expression patterns during larval development can illuminate the underlying mechanisms that drive this process.

The gene encoding the pituitary thyroid stimulating hormone (*tshb*) is strongly expressed very early on during larval development at D01, decreases from D03, and then strongly increases again at D32 (*Figure 3A*). Accordingly, we also observed two surges of expression for the hypothalamic factors *trhr1aa, trhr1a like, trhr1b,* and *trhr2,* at D03 and between D32 and D60, suggesting two distinct periods of stimulation of TH synthesis, one early on around D03 and one later at around D32. This pattern can also be seen in the expression of the corticotropin-releasing hormone (*crhb*) and receptors (*crhr1a, crhr1b,* and *crhr2*), which stimulates the synthesis of TH (*Denver, 2021*) (see section 'Possible involvement of corticoid pathways in metamorphosis' and Figure 6 below). Interestingly, we also observed a peak of expression for *tg* at D03, the gene encoding for the TH precursor, and a strong increase of expression starting at D32 (*Figure 3B*). The respective order of appearance of TSH and Tg (TSH at D32, Tg after) is consistent with what we would expect but a bit later than expected given the morphological transformation. It would be interesting to revisit this in a future series of experiments, with tighter temporal sampling to study how gene expression and morphological transformation aligned. A similar expression pattern was obtained for *tpo*, the gene encoding for the enzyme adding iodine to TH precursor, as well as for *sis*, the gene encoding for the symporter involved in transferring iodide into thyrocytes. Once produced, TH, particularly T4, are transported into target cells where they convert them into the active form T3 mostly by *dio2* and *dio1* or degraded by *dio3* and *dio1*. As it has been observed for HPT factors, *tg, tpo,* and *sis*, we first noticed two peaks of expression of *dio2* with a very early one at hatching (D01) and a second one at D32 suggesting two distinct periods in which active TH (that is T3) is required. In accordance with this observation, we notice a minimal expression of the T3 degrading enzyme *dio3* at these two periods followed by a final late increase after D32. *dio1*, whose net function is unclear (*Darras and Van Herck, 2012*), shows a regular increase of expression that becomes maximal at the juvenile stages (J) (*Figure 3C*). Finally, thyroid hormone receptors (TRs) expression levels increased throughout the entire larval development with a stronger increase of *trβ* at D60 (*Figure 3D*). Taken together, these data reinforce the existence of two distinct periods of TH signaling activity, one early on at D03, and one late at D32 (*Figure 3C*).

These results suggest the activation of the TH axis around D32, which coincides with the regression of the elongated appendages (second dorsal spine and pelvic spines) and the appearance of the adult-like pigmentation pattern, indicating that metamorphosis in *E. malabaricus* occurs around D32 in our rearing conditions. These observations are consistent with what has been observed in *E. coioides*, in which TH levels peak around 40 dph when the pelvic and second dorsal spines regress and adult-like pigmentation pattern formation is ongoing (*de Jesus et al., 1998*). Interestingly, the high expression levels of *tshb, trhr, tg, tpo, sis, dio3,* and TRs at the very beginning of development (D01-D03) suggest a precocious activation of TH synthesis, which, to our knowledge, has not been observed in groupers nor in other teleost fishes so far (*Figure 3*). Measurements of TH levels during these early development stages showed an early peak of T4 at D03, confirming the early activation of the TH pathway observed with gene expression patterns (*Figure 3E*).

## TH involvement in elongate appendage and regression

As mentioned in the introduction, many marine fish larvae present several morphological features that improve larval survival rates during their pelagic phase (*Miller and Kendall, 2019*). This is what we observe in grouper with the formation of elongated spines of the dorsal and pelvic fins that are supposed to have a defensive function (*Kawabe and Kohno, 2009*; *Cunha et al., 2013*; *Leu et al., 2005*). These spines then regress while adult-like pigmentation pattern appears and TH surge corresponding to the TH-regulated metamorphosis. It is well known that during fish larval development genes involved in ossification are under the controls of TH. In zebrafish, TH control the proper morphogenesis and ossification in the majority of the bones, during post-embryonic development and metamorphosis (*Keer et al., 2019*). This is why we investigated the expression changes of some of these genes in *E. malabaricus*. Interestingly, we observed, once again, two surges in the expression of the following genes: bone gamma-carboxyglutamate (*bglap*), periostin (*postnb*), and phosphate-regulating endopeptidase (*phex*), three key genes implicated in the mineralization of tissues. The first at D13 following the early surge in TH signaling genes, and the second starting at D60 (*Figure 4A*). The first surge of gene expression coincides with the appearance and growth of the dorsal and pelvic elongated spines starting at D10 (*Figure 4B*, shown by green arrowhead), while the second surge coincides with the regression of these spines, a process known to be regulated by TH in *E. coioides*

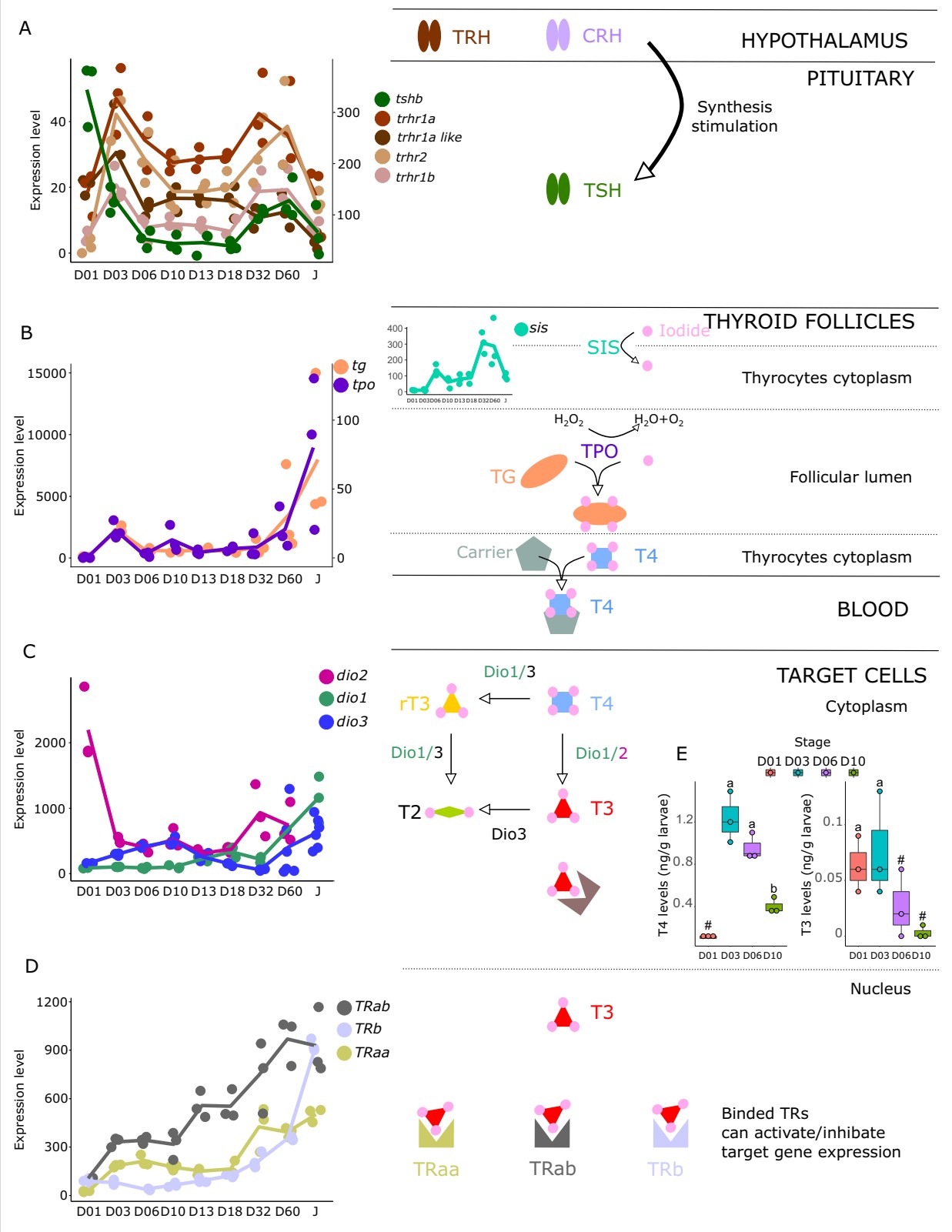

**Figure 3.** Expression levels of thyroid hormones (TH) signaling pathway genes in *E. malabaricus*. (**A**) TRH: thyroid releasing hormone, TSH: thyroid stimulating hormone. (**B**), DUOX: dual oxidase, TG: thyroglobulin, TPO: thyroperoxidase, SIS: sodium iodine symporter. (**C**) DIO: deiodinase. (**D**) TR: thyroid hormone receptor. Colored lines join the average values of each stage. Biochemical pathways adapted from ***Roux et al., 2023***. (**E**) T4 and T3 levels (in ng/g of larvae) during early larval development. Three biological replicates consisting of pooled larvae were analysed at each stage (D01 n =

*Figure 3 continued on next page*

*Figure 3 continued*

120 larvae per replicate, D03 n = 120 larvae per replicate, D06 n = 60 larvae per replicate, D10 n = 40 larvae per replicate). # indicates that the value is below the quantification limit, and different letters indicate significant differences <0.05 (one-way ANOVA followed by a Tukey HSD test for T4 levels only as no significant differences were observed after ANOVA for T3 levels). Gene expression data was generated from whole fish. Expression levels were derived from DESeq2 normalized gene counts.

The online version of this article includes the following source data for figure 3:

**Source data 1.** Raw thyroid hormone and cortisol measurements.

(*de Jesus et al., 1998*). The coincidence of both the growth and the regression of the elongated spines with the activation of the TH pathway in *E. malabaricus* may suggest that TH may play a role not only in the regression of these spines but also in their formation in this species.

## Other TH-regulated biological processes are also activated during grouper metamorphosis

Pigmentation changes are often the most visible changes in some teleost species such as clownfish (*Salis et al., 2021*). In grouper, the pigmentation changes are accompanied by the regression of the dorsal and pelvic spines. The acquisition of an adult pigmentation pattern is characterized by the formation of brown and white vertical bars in *E. malabaricus* (*Figure 4C*, juvenile stage). To reveal the molecular regulations driving these pigmentation changes, we assessed the expression of key pigmentation genes involved in white (iridophore genes), black (melanophore genes), and yellow (xanthophore genes) pigment cells known to be regulated by TH in zebrafish and clownfish (*Salis et al., 2021*; *Saunders et al., 2019*).

The expression level of the iridophore gene *flh2a* showed a strong increase from D03, followed by a decrease at D32 and a new surge at D60 (*Figure 4C*). The first increase may correspond to the appearance of iridophores on the ventral cavity whereas the second may coincide with the formation of the white bars. In contrast, its paralogue *fhl2b* remained relatively stable throughout the development. Xanthophores start colonizing the larval body at D10, which may explain the increase of the expression level of two xanthophores markers, *gtsm3*, and *perp6*, which play a role in concentrating and trafficking lipophilic pigments (*Granneman et al., 2017*). On the other hand, *scarb1*, which is involved in carotenoid deposition in zebrafish, increased slightly at D03 (*Figure 4C*). Similarly, melanophore genes are displaying a strong increase in their expression level at D03 and D06 that may be related to the colonization of melanophores on the larval body (*tyrp1b*, *tyr*, *Figure 4C*).

During their metamorphosis in the wild, fish larvae also undergo ecological changes such as habitat transition (from ocean to coastal environment) and food habits. It is well known that in many fish species, this ecological transition is accompanied by a change in color vision (*Cortesi et al., 2016*). Since TH appeared critical in the regulation of genes involved in vision in salmonids, zebrafish, and clownfish (*Roux et al., 2023*; *Volkov et al., 2020*; *Cheng et al., 2009*; *Veldhoen et al., 2006*), we investigated the regulation of genes encoding for visual opsin. We expected to find at least eight visual cone opsin genes in *E. malabaricus* according to the phylogeny of opsin genes in teleosts (*Cortesi et al., 2015*) (*opnsw1, opnsw2Aa, opnsw2Ab, opnsw2B, rh2A, rh2B, rh2C, opnlw*) and one rhodopsin gene (*rh1*) (*Cortesi et al., 2015*; *Musilova et al., 2021*). These genes were indeed expressed in our transcriptomic data. We observed that the medium wavelength opsin (*rh2A, rh2B, rh2C*), and the long wavelength opsin (*opnlw*) were highly expressed at the beginning of the larval development (at D03 for *rh2B, rh2C, and opnlw,* and at D10 for *rh2A*) (*Figure 4D*). These surges of expression are followed by the increase of the expression levels of *opnsw2B, opnsw2Bb, and opnlw* from D32. From D18 the rhodopsin involved in scotopic vision (*rh1*) increases. The expression level *opnsw1* remained low and stable during the entire development (*Figure 4D*). It is again very interesting to note that these changes coincide with both TH signaling peaks. As these genes are regulated by TH in other species and according to the observed expression patterns, we may assume that this is also the case in *E. malabaricus*.

The timing of cone opsin (*opnsw2a1, opsnw2a2, and opnlw*) expression in *E. malabaricus* is similar to *E. bruneus* (*Matsumoto and Ishibashi, 2016*), but different from *E. akaara* where *opnsw2* is strongly expressed early and then decreases (*Kim et al., 2019*). However, the expression levels of mid-wavelength opsins and *opnlw* are similar between *E. malabaricus* and *E. akaara*, suggesting

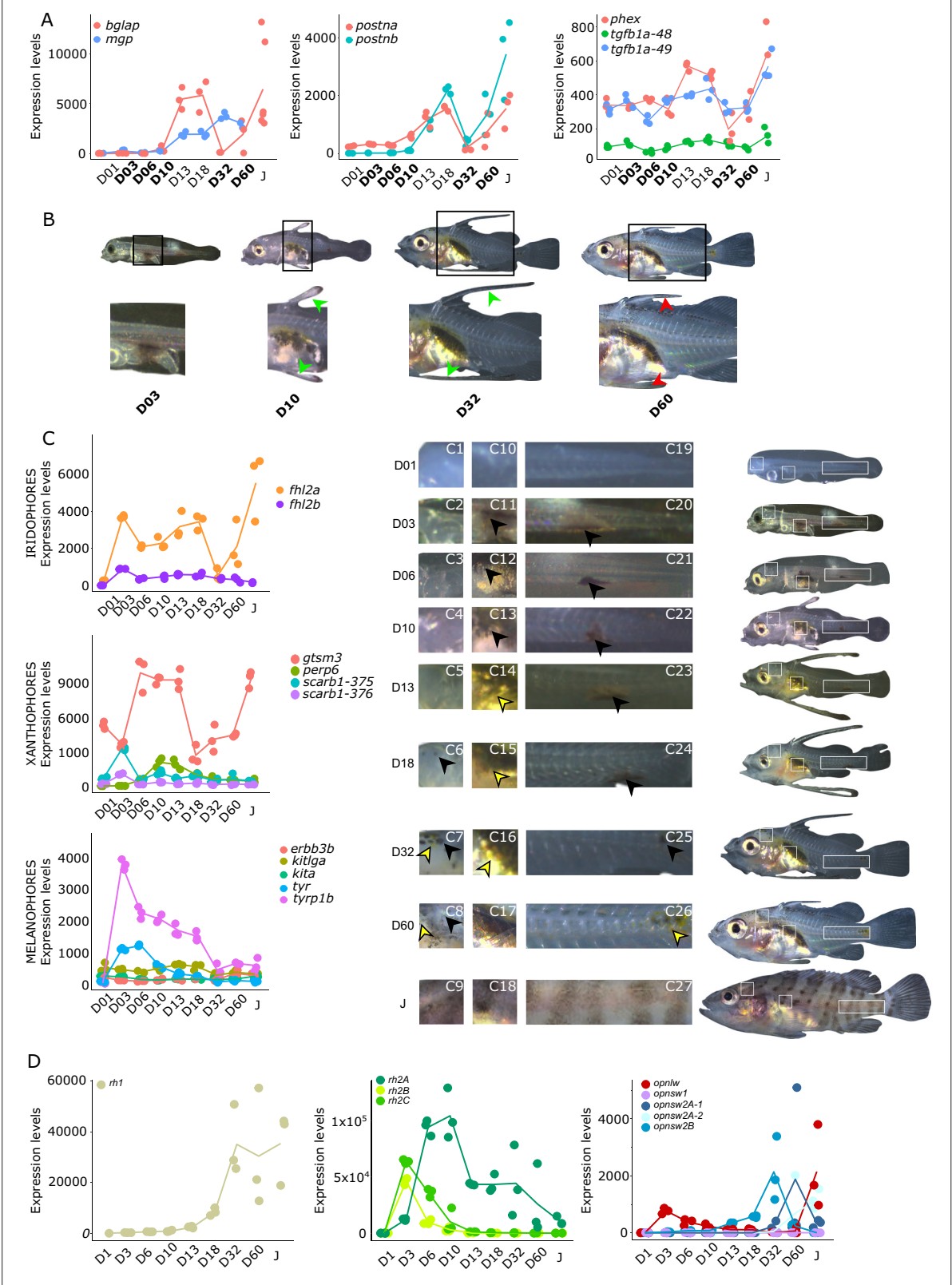

**Figure 4.** Biological processes likely to be under thyroid hormones (TH) control during *E. malabaricus* metamorphosis. (**A**) Expression patterns of key genes involved in the ossification process and known to be regulated by TH in teleosts. *Bglap*: bone gamma carboxyglutamate protein, *mgp*: matrix gla protein, *postna*: periostin a, *postnb*: periostine b, *phex*: phosphate regulating endopeptidase homolog X linked. (**B**) Pictures of *E. malabaricus* at D03, D10, D32, and D60 illustrate the elongation of the dorsal and pelvic floating spines (green arrow heads at D10 and D32) and their regression (red

*Figure 4 continued on next page*

*Figure 4 continued*

arrow heads at D60). (**C**) Expression patterns of genes involved in pigmentation. Three areas of interest were chosen to illustrate the appearance of melanophores (C6 to C8: at the top of the head, C11 to C13: above internal organs, and, C20 to C25: close to the caudal peduncle,) and xanthophores (C7 to C8 at the top of the head, C14 to C16 above internal organs and C26: close to the caudal peduncle). (**D**) Expression patterns of genes encoding for the rhodopsins (rh1) and the visual cone opsins (*rh2A, rh2B, rh2C, opnlw, opnsw1, opnsw2A-1, opnsw2A-2, opnsw2B*). Gene expression data was generated from whole fish. Expression levels were derived from DESeq2 normalized gene counts.

their involvement in cone photoreceptor differentiation, while rod photoreceptors differentiate during metamorphosis in *E. akaara* and *E. malabaricus* larvae.

## Metamorphosis is accompanied by a metabolic shift

Because metamorphosis is known to be energetically demanding and because the ecology of the planktonic larvae and the demersal juveniles are different, we investigated metabolic gene expression. *Figure 5* shows the expression profile of the genes encoding for the rate-limiting steps enzymes involved in glycolysis, (phosphofructokinase, *pfkma,* and *pfkmb*), and citric acid cycle (citrate synthase, *cs*; isocitrate dehydrogenase, *idh3a*; oxoglutarate dehydrogenase complex, *ogdhl, dlst2*). The expression profile of all the genes associated with these pathways are shown in *Figure 5—figure supplement 1*.

These profiles revealed a clear overall pattern: glycolysis genes are poorly expressed at the very beginning of the larval development while their expression increases throughout the development. This is particularly visible for *pfkma* which starts to increase from D10 and reaches its highest expression level at D32, likely coinciding with the onset of metamorphosis, and then decreases until the juvenile stage (J) (*Figure 5A*). The genes involved in the rate-limiting steps of the citric acid cycle (*cs,*

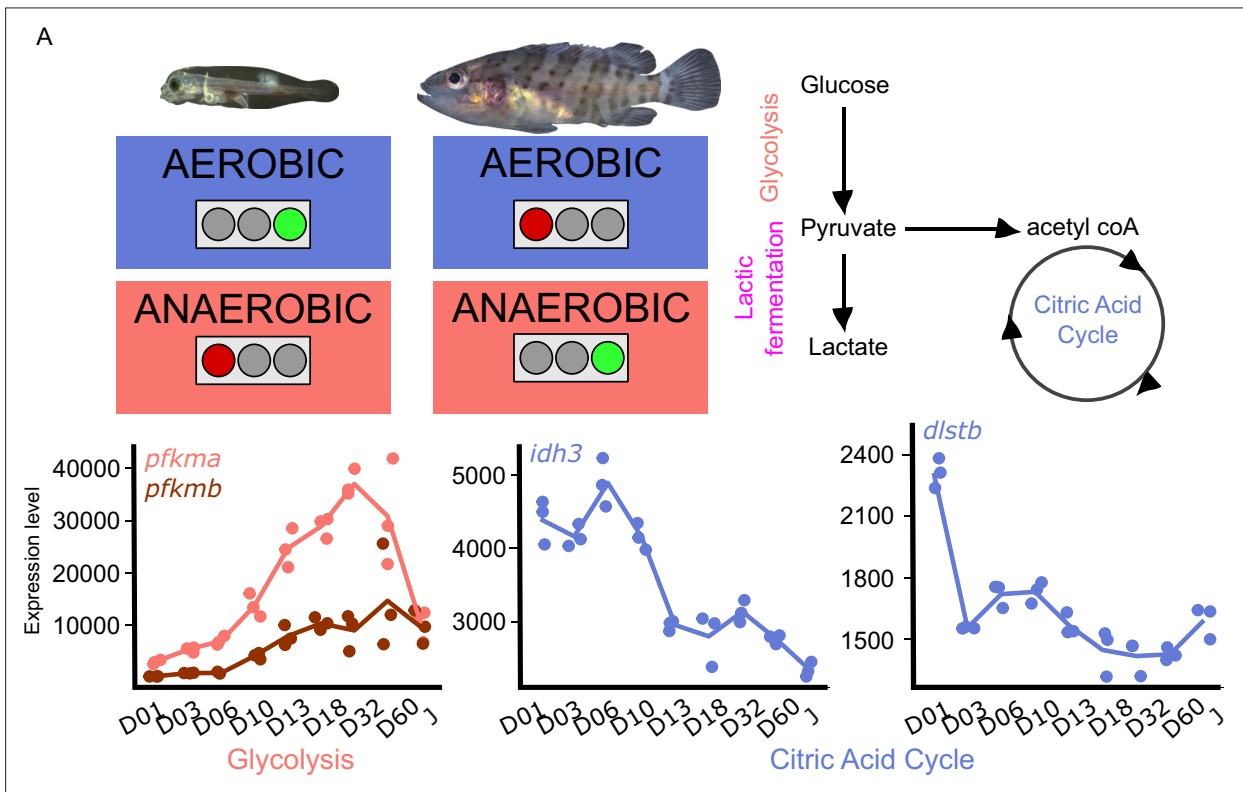

**Figure 5.** Metabolic transition and corticoid expression levels of *E. malabaricus*. (**A**) Schematization of the metabolic transition occuring during *E. malabaricus* larval development showing that young larvae rely on aerobic metabolism whereas older larvae rely on anaerobic metabolism. Expression levels of genes involved in glycolysis (*pfkma, pfkmb*), krebs cycle (*idh3, dlstb*). Gene expression data was generated from whole fish. Expression levels were derived from DESeq2 normalized gene counts.

The online version of this article includes the following figure supplement(s) for figure 5:

**Figure supplement 1.** Metabolic shift during grouper metamorphosis.

*idh3, dlst*) are more expressed during early larval stages and then decrease progressively. It is also worth noting that several genes involved in both glycolysis and the TCA cycle are encountering these two peaks of expression during the larval development (*gpi1b, aldoaa, gapdh1, pgam1a, pgam1b, pgam2, eno1b, pkma, dlsta, dldh, sdhb, mdh2*, Appendix 6). The lactic acid fermentation genes show an increase throughout the larval development with peaks of expression at D18 for *ldha* and at D03 for *ldhc* (**Figure 5—figure supplement 1**). Taken together, these results reveal that at the very beginning of the development larval fish mainly rely on the citric acid cycle for aerobic energy production and then switch progressively to anaerobic energy production via glycolysis and lactic fermentation. This trend is similar to what has been observed in other fish species such as sea bass (**Mazurais, 2011**; **Darias et al., 2008**), but contrasts with the situation of other species such as the clownfish (**Roux et al., 2023**). TH are known to play a role in the regulation of metabolism in mammals (**Mullur et al., 2014**), so it is likely that a similar regulatory process occurs during the development of *E. malabaricus* larvae, as it has been recently observed in the development of clownfish larvae (**Roux et al., 2023**). Larval development and metamorphosis are very sensitive periods during which larvae must face a myriad of challenges: disperse into the open ocean, find food, escape from predators, locate and swim toward a suitable habitat, metamorphose, and settle. All these challenges are highly demanding in terms of energy, it is thus very important for the larvae to properly allocate this energy to ensure the success of these various challenges. The regulation by TH of genes involved in processes such as glycolysis, lactic fermentation, and citric acid cycle might be a way for larvae to tune their energetic source to enhance their survival and the success of metamorphosis.

## Possible involvement of corticoid pathways in grouper larval development

Synergistic action of cortisol and THs has been encountered during flatfish larval development and more specifically during its metamorphosis. However, crosstalk between corticoids and TH pathways have remained poorly investigated during fish post-embryonic development (**Moster, 1981**). For this reason, we decided to investigate eight key genes genes involved in the Hypothalamo-Pituitary-Interrenal axis: *crha, crhb, crhr1a, crhr1b, crhr2, pomc-a1, pomc-a2, pomc-b, mr, gr1, gr2* which encodes, respectively, for the corticotropin-releasing hormone (which stimulates the production of POMC and the stress hormone ACTH), the receptors of the CRH which are involved in the production of the stress-related hormone ACTH the pro-opiomelanocortin A1, A2, and B (precursors of several hormones such as ACTH) and corticoid receptors: mineralocorticoid receptor (MR) and glucocorticoid receptor (GR1&2) (**Figure 6**) We also scrutinized the expression levels of genes encoding for key proteins involved in corticoid synthesis: *star, fdx1, fdx2, fdxr, cyp11a1, hsd3b1, cyp17a1, cyp21a2, cyp11c1, hsd11b1, hsd11b2*.

Most of the genes of this pathway displayed a similar pattern as described previously above with a surge of expression between D03 and D10 and a second one between D32, D60 (*crhb, crhr1b, crhr2, pomc-a2, mr, gr1*) (**Figure 6A**). The expression level of the gene encoding for CRHR2 started to increase after D01 and remained relatively stable all along whereas *crha* was lowly expressed (**Figure 6A**). A surge of expression was observed for *pomc-a1* at D03 followed by a constant decrease until the juvenile stage. High expression of *pomc-b* was observed at D13, D18, and Juvenile stage. Finally, *gr2* expression level increased strongly at D03, then remained stable and increased again at D60. The relatively high expression of the *crhr* genes may suggest an increase in the sensitivity to CRH to mediate the production of POMC by the pituitary gland, a process that seems to occur twice during *E. malabaricus* larval development.

Concomitantly, a two-step increase in the *star* is observed: first at D03 and a second at D32. This may suggest an increase in the production of cortisol following the high expression of *pomc-a2*. Indeed, POMC is the precursor of the adreno cortico trophic hormone (ACTH) which is the pituitary factor stimulating cortisol production by the inter-renal gland (**Takahashi and Mizusawa, 2013**). The expression levels of genes involved in cortisol production corroborate this hypothesis. Indeed, we observe an increase of expression around D03 and D06 for *fdx1, cyp11a1, hsd3b1, cyp11c1, hsd11b1, hsd11b2*, as well as a second increase of expression from D32 for *fdx1, fdxr, hsd3b1, cyp11c1, hsd11b1, hsd11b2*. Interestingly, measurements of cortisol levels during early larval development (between D01 and D10) showed that cortisol concentration starts to increase from D3, coinciding with the expression levels of the *star*, and is followed by a stronger increase from D10. Those results first indicate that the

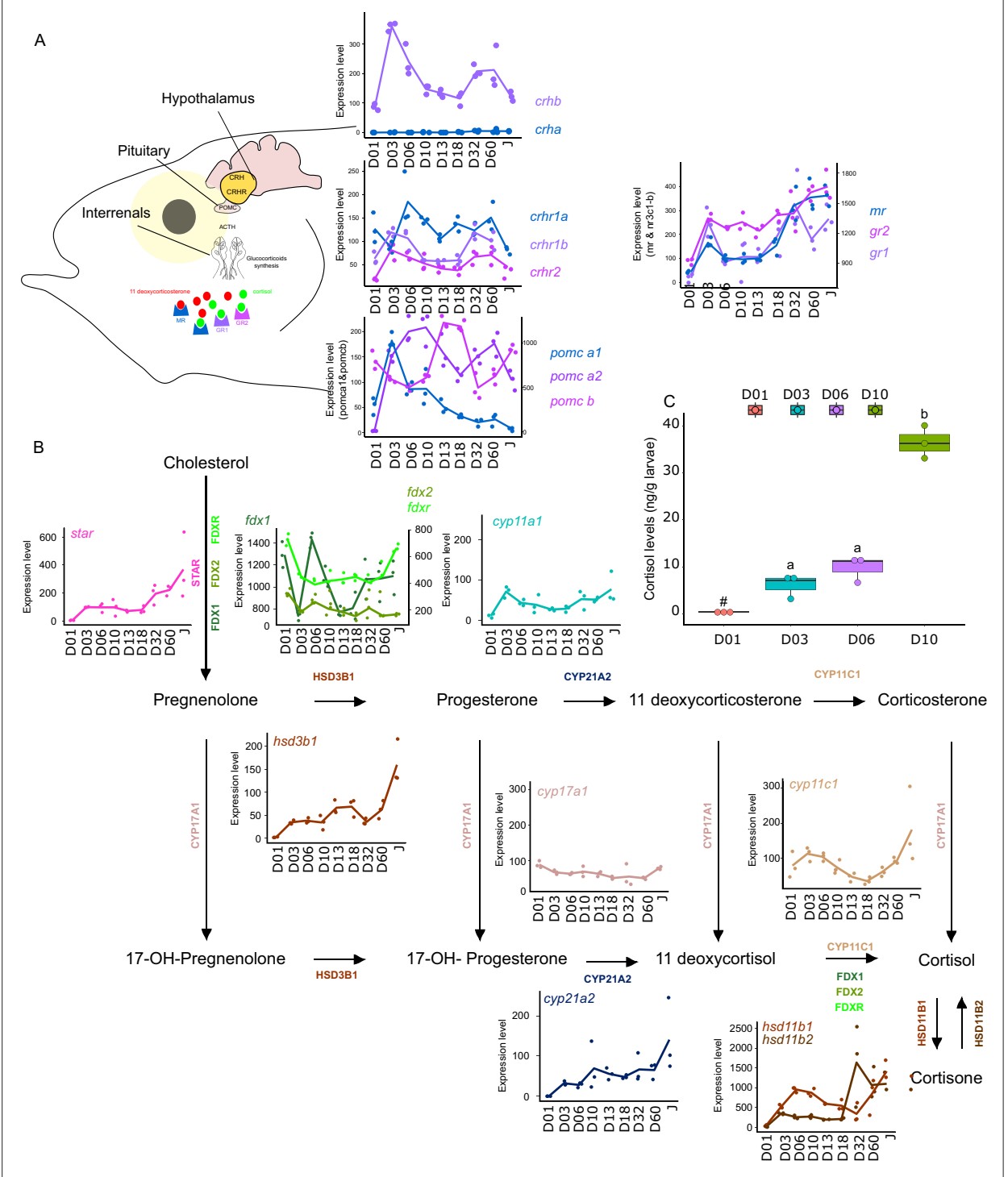

**Figure 6.** Expression levels of genes involved in the hypothalamo-pituitary-interrenal axis (HPI) and corticoid synthesis. (**A**) Expression levels of genes involved in HPI: *crha (corticotropin releasing hormone a)*, *crhb (corticotropin releasing hormone b)*, *crhr1a (corticotropin releasing hormone receptor 1 a)*, *crhr1b (corticotropin release hormone receptor 1b)*, *crhr2 (cortico release hormone receptor 2)*, *pomc-a1 (propiomelanocortin a1)*, *pomc-a2 (propiomelanocortin a2)*, *pomc-b (propiomelanocortin b)*, *mr (mineralocorticoid receptor)*, *gr1 (glucocorticoid receptor 1)*, *gr2 (glucocorticoid receptor 2)*. (**B**) Expression levels of genes involved in corticoids synthesis: *star (steroidogenic acute regulatory protein)*, *fdx1 (ferredoxine 1)*, *fdx2 (ferredoxine 2)*, *fdxr (ferredoxine reductase)*, *cyp11a1 (Cytochrome P450 Family 11 Subfamily A Member 1)*, *hsd3b1 (Hydroxysteroid dehydrogenases 3β1)*, *cyp17a1 (Cytochrome P450 Family 17 Subfamily A Member 1)*, *cyp21a2 (Cytochrome P450 Family 21 Subfamily A Member 2)*, *cyp11c1 (Cytochrome P450 Family 11 Subfamily C Member 1)*, *hsd11b1*, *hsd11b2 (Hydroxysteroid dehydrogenase 11 3β1&2)*. (**C**) Cortisol levels (in ng/g of larvae) during early larval development. Three biological replicates consisting of pooled larvae were analysed at each stage (D01 n = 120 larvae per replicate, D03 n = 120 larvae

*Figure 6 continued on next page*

*Figure 6 continued*

per replicate, D06 n = 60 larvae per replicate, D10 n = 40 larvae per replicate). # indicates that the value is below the quantification limit, and different letters indicate significant differences <0.05 (one-way ANOVA followed by a Tukey HSD test). Gene expression data was generated from whole fish. Expression levels were derived from DESeq2 normalized gene counts.

HPI axis and cortisol production are activated at the beginning of the larval development around the timing of activation of TH pathway genes between D03 and D10. Second, the transcriptomic data also showed an activation of the corticoid pathway genes around D32 as it has been observed for TH pathway genes. There is contrasting evidence of communication between these two pathways during teleost fish larval development with some data suggesting a synergic and other an antagonistic relationship. In terms of synergy, an increase in cortisol levels concomitantly with an increase in TH levels has been observed in flatfish (*de Jesus et al., 1991*), golden sea bream (*Deane and Woo, 2003*), and silver sea bream (*Szisch et al., 2005*). Cortisol was also shown to enhance in vitro the action of TH on fin ray resorption (a phenomenon occurring during flatfish metamorphosis) in flounder *de Jesus et al., 1990*. It has also been shown that cortisol regulates local T3 bioavailability in the juvenile sole via regulation of deiodinase 2 in an organ-specific manner (*Arjona et al., 2011*). On the antagonistic side, it has been shown that experimentally induced hyperthyroidism in common carp decreases cortisol levels (*Geven et al., 2006*), whereas cortisol exposure decreases TH levels in European eels (*Redding et al., 1986*). Given this scattered evidence, the existence of a crosstalk active during teleost larval development and metamorphosis has never been formally demonstrated. The results we obtained in grouper are clearly indicating that the HPI axis is activated during both early development and metamorphosis and that cortisol synthesis is activated during early development. This may suggest that in some aspects, cortisol synthesis could work in concert with TH, as has been shown in several different contexts in amphibians (*Sachs and Buchholz, 2019*), but functional experiments need to be conducted to confirm this hypothesis. It is worth to note, however, that the increase of the gene encoding POMC-A2 may not only be linked to cortisol synthesis as POMC is also a precursor of other hormones and notably melanocytes-stimulating hormones (*Takahashi and Mizusawa, 2013*). Those hormones belong to the melanocortin system that is involved in body pigmentation, but also in social behavior, appetite, and stress physiology (*Cone, 2006*). The increase in *pomc-a2* observed during *E. malabaricus* may thus also be involved in the onset of pigmentation pattern. Taken together, these results brought a first insight into the potential role of corticoids in the larval development of *E. malabaricus* and call for functional experiments directly testing a possible synergy. Given the results obtained in our study, *E. malabaricus* could be a good model to investigate the potential role of corticoids and TH in elongate appendages formation during early larval development as well as during metamorphosis and if there is an interplay between the two pathways. Such interplay could have relevant consequences in terms of aquaculture and claim for an examination of the role of stress in regulating fish larval development and impacting metamorphosis triggering.

Overall, the results obtained in this study revealed a very precocious surge of expression of genes involved in two key hormonal pathways (corticoids and TH) that are known to control ontogenetic transitions, but which are also involved in the regulation of many biological processes (*Wada, 2008*; *Watanabe et al., 2016*). This indicates that the early post-embryonic period in grouper may correspond to such an ontogenetic transition that has been ignored until now and that could be linked to the formation of the specific elongated appendages present in groupers.

More generally, the fact that the outcome of metamorphosis is very variable from one species to another (e.g. differences in metamorphosis between clownfish, grouper, flatfish, etc.) and that it also allows exquisite acclimation of the juveniles to their local environment (*Denver, 2021*), highlights the capacity of this transitional step, controlled by environmentally connected hormonal systems, to change rapidly in accordance with ecological needs (*Zwahlen et al., 2024*). Finally, considering that rearing conditions during larval metamorphosis in an aquaculture context may impact growth and welfare at later life stages, understanding the molecular changes occurring during the development of a species might prove useful to enhance survival rates.

# Materials and methods

## Larval husbandry

This study was conducted in partnership with the Okinawa Prefectural Sea Farming Center, Motobu-cho, Okinawa, Japan. *Epinephelus malabaricus* larvae and juveniles were obtained from various clutches obtained from natural spawning in 2020, 2021, and 2023. Larvae were reared under natural conditions in 50,000 L of natural sea water in circular tanks. Light exposure duration followed natural daylight hours, salinity (approximately 33–34 ppm), and temperature (approximately 27°C on average) remained relatively stable as the tanks were constantly renewed with natural seawater. Microalgae (*Nannochloropsis* sp.) was added from hatching until 15 days post-hatching (dph) to maintain the nutritional value of live-feed organisms and create a green-water environment. Rotifers *Brachionus* sp. (S type) were enriched with fish oil and distributed twice a day from 1 dph to maintain a concentration of 10 ind/mL until 13 dph. Artemia nauplii were added twice a day from 13 dph to 20 dph. Frozen copepods were given five times a day from 13 dph until 20 dph. Artificial food was given from 20 dph during the daytime by automatic feeding (one distribution every hour).

## Sample collection and tissue collection

In order to assemble and functionally annotate the genome, tissues for DNA sequencing and RNA sequencing were collected on September 8, 2020 from two approximately 4-month-old fish sourced from the Okinawa Prefectural Sea Farming Center. The fish were euthanized by cervical dislocation, and immediately dissected. The liver and muscle tissues of one fish were immediately frozen in liquid nitrogen for PacBio HiFi and Hi-C sequencing, respectively. Brain, gill, liver, heart, caudal fin, eye, spleen, stomach, intestine, muscle, skin spinal cord, and spinal nerve tissues were taken from the second fish and stored in RNAlater (ThermoFisher Scientific) for tissue-specific transcriptome sequencing.

For the larval developmental analysis, whole larval and juvenile fish were sampled between April 30, 2021 and June 2, 2021, ranging from 1 day post-hatching (dph) to approximately 2 months (*Table 4*). A total of four clutches spawned in early and late April were sampled during this period and larvae were collected and sorted according to their morphology allowing us to sequence eight developmental stages. Larvae and juveniles were euthanized in the afternoon (between 13:00 and 15:00) with MS222 solution (200 mg/L, Sigma-A5040) before being placed in RNAlater. Larger fish were cut open for improved RNAlater penetration and samples were kept at 4°C for 2-8 days before being stored at –20°C until extraction. Larvae for TH and cortisol measurements were sampled in triplicates between June 17, 2023 and June 26, 2023 at D01 (n=120 per replicate), D03 (n=120 per replicate), D06 (n=60 per replicate), and D10 (n=40 per replicate), as described in *Roux et al., 2023* and kept at –80 until analysis. TH and cortisol extraction and measurement were outsourced to ASKA Pharmaceutical Medical Co., Kanagawa, Japan. Detailed protocols can be found in Appendix 1 for TH and Appendix 2 for cortisol.

All sampling conducted in this study was done under the approval of the Animal Care and Use Committee at the Okinawa Institute of Science and Technology Graduate University (approval N°2021–328).

## DNA extraction and sequencing

Genomic DNA was extracted from liver tissue using the NucleoBond HMW DNA extraction kit (Machery-Nagel). Library preparation was carried out with the SMRTbell Express Template Prep Kit 2.0 and SMRTbell Enzyme Cleanup Kit, Sequencing primer v2, Sequel II Binding Kit 2.0, and Sequel II Sequencing Kit 2.0 (Pacific Biosciences). Sequencing was done on a Sequel II System, using three SMRT Cell 8 M flow cells through diffusion loading of 60-100pM library. Hi-C library preparation and sequencing was carried out by Phase Genomics from muscle tissue using the Phase Genomics Proximo Animal Kit v3.0 and sequenced on a Illumina HiSeq 4000 with 150 bp PE.

## RNA extraction and sequencing

For the functional genome annotation, tissue samples were homogenized using a Kinematica Polytron PT1200E Homogenizer and RNA was extracted using the Maxwell RSC simply RNA Tissue Kit (Promega: AS1340). Individually barcoded IsoSeq Express libraries of all 13 tissues were prepared by

**Table 4.** Morphological description of the larval and juvenile stages sampled for the transcriptomic analysis.

D01: 1 day post hatching (dph), D03: 3 dph, D10: 10 dph, D:13 13–15 dph, D18: 18–20 dph, D32: 32–34 dph, D60: ca. 60 dph, J: ca. 60 dph with juvenile phenotype. NL is "notochord length" for preflexion and flexion larvae, SL is "standard length" for postflexion and older stages, and TL is "total length" for all stages.

| Age (dph) | | Timpoint/ Stage | Morphological description |
|---|---|---|---|
| 1 | 2.5 mm NL/2.7 mm TL | D01 | Newly hatched larva with a yolk sac; mouth unopened; eyes not pigmented; no pectoral fin |
| 3 | 2.7 mm NL/2.9 mm TL | D03 | Yolk sac remains; the mouth is opened; eyes are pigmented; pectoral fins are formed; large melanophores appear on the ventral cavity and on the second half of the body |
| 6 | 2.9 mm NL/3.1 mm TL | D06 | Yolk sac has been resorbed; dorsal-fin spine starts to form within the fin fold |
| 10 | 3.8 mm NL/4.0 mm TL | D10 | Embryonic fin fold start differentiating in anal and dorsal fin while second spine of dorsal fin and spines of pelvic fins begin to extend with some melanophores colonizing the tips and xanthophores start covering the ventral cavity |
| 13–15 | 6.0 mm NL/6.4 mm TL | D13 | Spines of dorsal and pelvic fins grow. First spine of dorsal fin appears, second spine of dorsal fin and spines of pelvic fins become serrated; head spines appear, caudal-fin rays start to form, tip of the notochord begins to flex; xanthophores continue their expansion |
| 18–20 | 6.8 mm SL/8.3 mm TL | D18 | Notochord post-flexed; hypural bones are formed and in perpendicular position; caudal-fin rays are segmented; soft rays of dorsal and anal fins start to form and both fins start to form their final shape; fin rays are forming on upper part of the pectoral fin; soft rays of pelvic fins began to form; melanophores appeared on the top of the head and on the caudal peduncle |
| 30–32 | 10.0 mm SL/12.8 mm TL | D32 | Second spine of dorsal fin and spines of pelvic fins start to regress; soft rays in dorsal, anal, and pectoral fins are weakly segmented, caudal fin becomes truncated shape; melanophores are appearing at the basis of dorsal spines and along the notochord, melanophores ventrally on the caudal peduncle disappears; xanthophores start colonizing the caudal peduncle |
| 60 | 14.7 mm SL/19.1 mm TL | D60 | Second spine of dorsal fin and spines of pelvic fins continue their regression. Soft rays in pectoral and pelvic fins segmented; caudal-fin rays branched; anterior two bands of melanophores start appearing in some individuals: xanthophores are disappearing from the ventral cavity |
| 60 | 27.6 mm SL/34.6 mm TL | J | Juvenile stage; scales cover the body surface; second spine of dorsal fin and spines of pelvic fins fully regressed and became plain without hooks; caudal fin reached its final round shape; adult pigmentation pattern is more visible with alternate light and brownish vertical bands making lateral line system fully visible |

the OIST Sequencing Section using the SMRTbell Express Template Prep Kit 2.0. The libraries were sequenced on a PacBio Sequel 2 across two SMRT Cell 8 M flow cells.

For the developmental transcriptomic analysis, samples from 1 to 32 dph were homogenized in thioglycerol using metal beads lysing matrix tubes (MPB) in an automated homogenizer (FastPrep-24 5 G MPB). Bigger samples (60 dph and juveniles) were manually homogenized in thioglycerol using 14 mL round bottom tubes and a tissue grinder (Tissue Ruptor II, Qiagen). Samples from 1 and 3 dph consisted of pools of three larvae in triplicates, while all remaining timepoints consisted of triplicates of single individuals. RNA extraction was then carried out as for the tissue samples using the Maxwell

**Table 5.** Origin of protein sequences used for genome annotation in braker2.

| Species | Common name | Number of proteins | Source |
|---|---|---|---|
| *Amphiprion ocellaris* | Ocellaris clownfish | 48,668 | https://www.ncbi.nlm.nih.gov/protein |
| *Danio rerio* | zebrafish | 88,631 | https://www.ncbi.nlm.nih.gov/protein |
| *Acanthochromis polyacanthus* | spiny chromis damselfish | 36,648 | https://www.ncbi.nlm.nih.gov/protein |
| *Oreochromis niloticus* | Nile tilapia | 36,648 | https://www.ncbi.nlm.nih.gov/protein |
| *Oryzias latipes* | Japanese medaka | 47,623 | https://www.ncbi.nlm.nih.gov/protein |
| *Poecilia reticulata* | guppy | 45,692 | https://www.ncbi.nlm.nih.gov/protein |
| *Salmo salar* | Atlantic salmon | 112,302 | https://www.ncbi.nlm.nih.gov/protein |
| *Stegastes partitus* | bicolor damselfish | 31,760 | https://www.ncbi.nlm.nih.gov/protein |
| *Takifugu rubripes* | Japanese puffer | 49,529 | https://www.ncbi.nlm.nih.gov/protein |
| *Epinephelus lanceolatus* | Giant grouper | 42,970 | GCA_005281545.1, RefSeq |
| *Epinephelus akaara* | Red-spotted grouper | 23,923 | 4398b9f, Dryad |
| Total aa sequences | | 1,155,478 | |

RSC simply RNA Tissue Kit (Promega: AS1340). Library preparation was carried out at the OIST Sequencing Section using the NEBNext Ultra II Directional RNA Library Prep Kit. The final pooled library was then split into two Illumina Nova Seq SP flowcells for sequencing with 150 bp PE reads.

## Genome assembly, scaffolding, and phasing

The genome assembly was carried out using unprocessed PacBio HiFi reads with the diploid aware Improved Phased Assembler (https://github.com/PacificBiosciences/pbipa; *Sović and Kronenberg, 2020*) using default parameters, which resulted in a primary and alternative phase genome. The two-phased genomes were assessed using purge_haplotigs (*Roach et al., 2018*) using default parameters to generate a genome-wide read-depth histogram; however, no purging was necessary. Completeness of the final assembly was assessed using BUSCO (V4.1.2) (*Manni et al., 2021*) with the actinopterygii_odb10 database. Scaffolding and phasing were outsourced to Phase Genomics (See Appendix 3 for details).

## Genome and functional annotation

Genome annotation was carried out as described (*Ryu et al., 2022*). Briefly, repeat content analysis was done in RepeatModeler (*Flynn et al., 2020*) (V2.0.1), RepeatMasker (*Tempel, 2012*) (V4.1.1), the vertebrata library of Dfam (V3.3) (*Storer et al., 2021*), and GenomeTools (V1.6.1) (*Gremme et al., 2013*). Annotation was done using BRAKER2 (*Brůna et al., 2021*) and associated programs (*Barnett et al., 2011*; *Brůna et al., 2020*; *Buchfink et al., 2015*; *Gotoh, 2008*; *Hoff, 2019*; *Hoff et al., 2016*; *Iwata and Gotoh, 2012*; *Li et al., 2009*; *Lomsadze et al., 2014*; *Lomsadze et al., 2005*; *Stanke et al., 2008*; *Stanke et al., 2006*). For this, the ISO-seq data from the adult tissue and RNA-seq data from the larval samples (see below for the quality control process) were used together with publicly available protein data (*Table 5*). Post-processing was carried out as described by *Ryu et al., 2022* using the Swiss-Prot protein database (UniProt) (*Consortium, 2021*) with Diamond (*Buchfink et al., 2015*) (V2.0.9) and Pfam domains (*Mistry et al., 2021*) identified by InterProScan (V5.48.83.0) (*Zdobnov and Apweiler, 2001*). Gene model statistics were calculated using the get_general_stats. pl script from the eval package (V2.2.8) (*Keibler and Brent, 2003*). Finally, functional annotation was

carried out with the filtered gene models produced by BRAKER. The amino acid sequences were blasted against the non-redundant protein database (downloaded 15. November 2021) using blastp (V2.10.0+; parameters: -show_gis -num_threads 10 -evalue 1e-5 -word_size 3 -num_alignments 20 -outfmt 14 -max_hsps 20) (*Altschul et al., 1990*). Additionally, protein domains were assigned using InterProScan (V5.48.83.0; parameters: --disable-precalc --goterms --pathways -f xml) (*Zdobnov and Apweiler, 2001*). The blast and interproscan results were then loaded into OmicsBox (*Gotz et al., 2008*; *Huerta-Cepas et al., 2017*) for post-processing.

### Differential gene expression analysis

The differential gene expression analysis for the larval developmental stages was carried out on the sequencing data from the whole larval and juvenile fish. Before processing, the data from the two lanes were merged per sample. Low-quality bases and adaptor sequences were filtered using Trim Galore (V0.6.5) (*Krueger, 2015*) and cutadapt (V2.10) (*Martin, 2011*) using default parameters with the exception of '--length 30.' Kraken2 (V2.0.9-beta) (*Wood et al., 2019*) was used to remove bacterial reads using the bacterial and archeal database (V4.08.20) and '--confidence 0.3.' Cleaned reads were mapped using STAR (V2.7.9a) (*Dobin et al., 2013*) with '--quantMode GeneCounts' and '--outSAMtype BAM SortedByCoordinate,' using the filtered gff file produced by the braker2 annotation outlined above for the genome indexing (--genomeSAindexNbases 13, --sjdbOverhang 149). The unstranded mapped reads were then loaded into Rstudio (V2022.02.4) (*Team, 2020*) using R (V3.6.3) (*R Development Core Team, 2013*). DESeq2 (V1.36.0) (*Love et al., 2014*) was used for general data analysis, with coseq (V1.20.0) (*Godichon-Baggioni et al., 2019*; *Rau and Maugis-Rabusseau, 2018*) being used for cluster analysis. The cluster analysis was carried out on differentially expressed genes only, as determined through likelihood ratio test (LRT) analysis (full model: design = ~dph, reduced model: reduced = ~1, adjusted p-value threshold: 0.001) in DESeq2. Adjusted p-values and annotations for the group of genes represented in *Figure 1B* in this study can be found in the Suppl. Data File. Normalization was done in DESeq2, while the following parameters were used for coseq: model = 'Normal,' transformation = 'arcsin,' seed = 1234, iter = 10,000. Specific genes belonging to clusters where D03 and/or Day 32 showed upregulation and were then re-clustered with the same parameter for visualization. A complete representation of all initial clusters found in this study can be found in *Figure 2—figure supplement 1*. Pairwise analysis of differentially expressed genes between two-time points was done using the Wald test in DESeq2 (design = ~dph, adjusted p-value threshold: 0.01, log2FoldChange ≥ ±0.58). Figures were plotted using ggplot2 (V3.4.1) (*Wickham, 2009*), and the analysis made general use of the tidyverse package (V1.3.2) (*Wickham et al., 2019*). Lastly, expression levels shown in *Figures 3–6* are normalized gene counts produced by DESeq2.

### Materials and correspondence

Correspondence and material requests should be addressed to Roger Huerlimann at either roger. huerlimann@oist.jp or roger.huerlimann@gmail.com.

## Acknowledgements

We are grateful to Misaki Yamauchi and Hiroyuki Nakamura at the Okinawa Prefectural Sea Farming Center, who provided us with the samples of groupers. This work was supported by the Okinawa Institute of Science and Technology Graduate University. We are also thankful for the help and support provided by the Sequencing Section, especially Mayumi Kawamitsu and Albert Murzabaev, and the Scientific Computing Section of the Research Support Division at Okinawa Institute of Science and Technology Graduate University. We thank Konstantin Khalturin (Marine Genomics Unit, OIST) for his support of our sampling. Scaffolding was carried out by Mary Wood from Phase Genomics. We are grateful to Erina Kawai for her assistance in sourcing the fish. Vincent Laudet would like to dedicate this manuscript to the memory of Donald D Brown. No external funding was received for this work.

## Additional information

### Funding
No external funding was received for this work.

### Author contributions
Roger Huerlimann, Conceptualization, Data curation, Formal analysis, Supervision, Investigation, Visualization, Methodology, Writing – original draft, Project administration, Writing – review and editing; Natacha Roux, Formal analysis, Investigation, Visualization, Methodology, Writing – original draft, Writing – review and editing; Ken Maeda, Resources, Investigation, Visualization; Polina Pilieva, Investigation, Visualization; Saori Miura, Investigation, Project administration; Hsiao-chian Chen, Michael Izumiyama, Investigation; Vincent Laudet, Conceptualization, Supervision, Methodology, Writing – review and editing; Timothy Ravasi, Conceptualization, Supervision, Funding acquisition, Writing – review and editing

### Author ORCIDs
Roger Huerlimann (ID) https://orcid.org/0000-0002-6020-334X
Natacha Roux (ID) http://orcid.org/0000-0002-1883-7728
Ken Maeda (ID) https://orcid.org/0000-0003-3631-811X
Saori Miura (ID) http://orcid.org/0000-0002-2017-1697
Michael Izumiyama (ID) http://orcid.org/0000-0002-5886-1415
Vincent Laudet (ID) http://orcid.org/0000-0003-4022-4175
Timothy Ravasi (ID) http://orcid.org/0000-0002-9950-465X

### Ethics
All sampling conducted in this study was done under the approval from the Animal Care and Use Committee at the Okinawa Institute of Science and Technology Graduate University (approval N°2021-328).

Reviewer #1 (Public Review): https://doi.org/10.7554/eLife.94573.3.sa1
Author response https://doi.org/10.7554/eLife.94573.3.sa2

## Additional files

### Supplementary files
• Supplementary file 1. Differentially expressed, upregulated, and downregulated genes between consecutive time points.

• MDAR checklist

• Source data 1. Raw gene count matrix of larval stages mapped against the malabar grouper genome.

• Source data 2. Annotation of significantly differentially expressed genes which are upregulated at both D03 an D32.

### Data availability
Code used in genome assembly and annotation, as well as in developmental transcriptome analysis can be found under the following DOI: https://doi.org/10.5281/zenodo.10972118. All raw and assembled data used in this study has been deposited on GenBank under umbrella BioProject PRJNA798702, with the principal phased assembly in BioProject PRJNA798188, the alternate phased assembly in BioProject PRJNA798189, and the raw data in BioProject PRJNA794870. BioSamples can be found under SAMN24662200 (genome sequencing), SAMN24664212 (ISO-seq), and SAMN24664213 - SAMN24664234 / SAMN32359227 - SAMN32359229 (RNA-seq). PacBio and Illumina raw data can be found under SRR17639994 - SRR17640023 / SRR22859365 - SRR22859367. The final scaffolded assembly can be found under accession JANUFT000000000 and the alternative phase under accession JANUFU000000000. The genome and functional annotation are deposited on figshare under https://doi.org/10.6084/m9.figshare.25486387.v2 .The raw gene expression count data can be found in Source Data 1.

The following datasets were generated:

| Author(s) | Year | Dataset title | Dataset URL | Database and Identifier |
|---|---|---|---|---|
| Huerlimann R | 2024 | Chromosome level assembly of the malabar grouper (Epinephelus malabaricus genome) genome | https://www.ncbi.nlm.nih.gov/bioproject/?term=PRJNA798702 | NCBI BioProject, PRJNA798702 |
| Huerlimann R | 2024 | Epinephelus malabaricus (Malabar grouper) | https://www.ncbi.nlm.nih.gov/bioproject/?term=PRJNA794870 | NCBI BioProject, PRJNA794870 |
| Huerlimann R | 2024 | Malabar grouper (Epinephelus malabaricus) genome resources | https://doi.org/10.6084/m9.figshare.25486387.v2 | figshare, 10.6084/m9.figshare.25486387.v2 |

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

# Appendix 1

## T3 and T4 measurements

ASKA Pharmaceutical Medical Co., Ltd., 5-36-1 Shimosakunobe, Kawasaki Takatsu-ku, Kanagawa 213–8522, Japan.

Before sample preparation, a whole fish body was homogenized in distilled water by a ball mill (ShakeMaster NEO) with stainless beads. The homogenate sample was transferred to a polypropylene (PP) tube and spiked with isotope-labelled internal standards solution containing $^{13}C_6$-T4, $^{13}C_6$-T3, and $^{13}C_6$-rT3. The homogenate sample was denatured with acetonitrile and then, equilibrated for 30 min at room temperature on dark

After equilibration, the sample was centrifuged at 3000 rpm for 3 min, and then the supernatant was decanted into a new PP tube which was added to distilled water. The sample was applied to an Oasis MCX cartridge which had been successively conditioned in methanol, distilled water, and 1% acetic acid solution. After the cartridge was washed with distilled water followed by methanol, the thyroid hormones were eluted with methanol/distilled water/ammonia solution (70:30:1,v/v/v). After the sample was evaporated to dryness, the residue was dissolved with methanol/distilled water pyridine solution (40:60:1, v/v/v). The sample was subjected to an LC-MS/MS system for determination of T4, T3, and rT3. The SRM transitions were *m/z* 777.8/731.6 for T4, 651.9/605.8 for T3 and 651.9/507.7 for rT3. The measurement ranges were 4–4000 pg/tube for T4 and 0.5–500 pg/tube for both T3 and rT3. The limits of quantification were 4 pg/tube for T4 and 0.5 pg/tube for both T3 and rT3.

## Appendix 2

### Cortisol measurements

ASKA Pharmaceutical Medical Co., Ltd., 5-36-1 Shimosakunobe, Kawasaki Takatsu-ku, Kanagawa 213–8522, Japan

Before sample preparation, a whole fish body was homogenized in distilled water by a ball mill (ShakeMaster NEO) with stainless beads. The homogenate sample was transferred to a glass tube and spiked with an isotope-labelled internal standard solution containing F-d4. F was extracted with 4 mL of methyl *tert*-butyl ether.

After the organic layer was evaporated to dryness, the extract was dissolved in 0.5 mL of methanol and diluted with 1 mL of distilled water. The sample was applied to an OASIS MAX cartridge which had been successively conditioned with 3 mL of methanol and 3 mL of distilled water. After the cartridge was washed with 1 mL of distilled water, 1 mL of methanol/distilled water/acetic acid (45:55:1,v/v/v), and 1 mL of 1% pyridine solution, the F was eluted with 1 mL of methanol/pyridine (100:1,v/v).

After evaporation, the residue was reacted with 50 µL of mixed solution (80 mg of 2-methyl-6-nitrobenzoic anhydride, 20 mg of 4-dimethylaminopyridine, 40 mg of picolinic acid and 10 µL of triethylamine in 1 mL of acetonitrile) for 30 min. at room temperature. After the reaction, the sample was dissolved in 0.5 mL of ethyl acetate/hexane/acetic acid (15:35:1, v/v) and the mixture was applied to a HyperSep SI cartridge which had been successively conditioned with 3 mL of acetone and 3 mL of hexane. The cartridge was washed with 1 mL of hexane, and 2 mL of ethyl acetate/hexane (3:7, v/v). Steroids was eluted with 2.5 mL of acetone/hexane (7:3, v/v). After evaporation, the residue was dissolved in 0.1 mL of acetonitrile/distilled water (2:3, v/v), and the solution was subjected to an LC-MS/MS. The SRM transitions was $m/z$ 468.2/309.2 for F and 472.2/454.3 for F-d4. The measurement range was 10–100000 pg/tube. The limit of quantification of F was 10 pg/tube.

## Appendix 3

### HiC scaffolding protocol used by phase genomics

Chromatin conformation capture data was generated using a Phase Genomics (Seattle, WA) Proximo Hi-C 4.0 Kit, which is a commercially available version of the Hi-C protocol (*Lieberman-Aiden et al., 2009*). Following the manufacturer's instructions for the kit, intact cells from two samples were crosslinked using a formaldehyde solution, digested using the DPNII restriction enzyme, and proximity ligated with biotinylated nucleotides to create chimeric molecules composed of fragments from different regions of the genome that were physically proximal in vivo, but not necessarily genomically proximal. Continuing with the manufacturer's protocol, molecules were pulled down with streptavidin beads and processed into an Illumina-compatible sequencing library. Sequencing was performed on an Illumina HiSeq 4000, generating a total of 159,606,973 read pairs.

Reads were aligned to the draft assembly also following the manufacturer's recommendations (*Phase Genomics, 2024*). Briefly, reads were aligned using BWA-MEM (*Li and Durbin, 2010*) with the –5SP and -t 8 options specified, and all other options default. SAMBLASTER (*Faust and Hall, 2014*) was used to flag PCR duplicates, of which were later excluded from the analysis. Alignments were then filtered with samtools (*Li et al., 2009*) using the -F 2304 filtering flag to remove non-primary and secondary alignments. FALCON-Phase (*Kronenberg et al., 2018*) was used to correct likely phase switching errors in the primary contigs and alternate haplotigs from FALCON-Unzip and output its results in pseudohap format, creating one complete set of contigs for each phase.

Phase Genomics' Proximo Hi-C genome scaffolding platform was used to create chromosome-scale scaffolds from FALCON-Phase's phase 0 assembly, following the same single-phase scaffolding procedure described in *Bickhart et al., 2017*. As in the LACHESIS method (*Burton et al., 2013*), this process computes a contact frequency matrix from the aligned Hi-C read pairs, normalized by the number of DPNII restriction sites (GATC) on each contig, and constructs scaffolds in such a way as to optimize expected contact frequency and other statistical patterns in Hi-C data.

Juicebox (*Durand et al., 2016*; *Rao et al., 2014*) was then used to correct scaffolding errors, and FALCON-Phase was run a second time to detect and correct phase switching errors that were not detectable at the contig level, but which were detectable at the chromosome-scaffold level. Metadata generated by FALCON-Phase about scaffold phasing was used to generate matching. assembly files (a file format used by Juicebox) and subsequently used to produce a diploid, fully-phased, chromosome-scale set of scaffolds using a purpose-built script (*Sullivan, 2021*). In these final scaffolds, phase 0 included 24 scaffolds spanning 1,027,591,625 bp (92.82% of input) with a scaffold N50 of 43,313,630 bp, and phase 1 included 24 scaffolds spanning 1,019,278,383 bp (91.98% of input) with a scaffold N50 of 43,164,609 bp.

