## [Editor Report · eLife assessment]

The work provides **valuable** genomic resources to address the endocrine control of a life cycle transition in the Malabar grouper fish. The revised manuscript is more **solid** and the resources and experimental data help to build up a meaningful biological understanding of thyroid signaling in grouper fish.

---

## [Referee Report · Reviewer #1 (Public Review)]

Summary and strength:

The authors undertook to assemble and annotate the genome sequence of the Malabar grouper fish, with the aim to provide molecular resources for fundamental and applied research. Even though this is more mainstream, the task is still daunting and labor intensive. Currently, high quality and fully annotated genome sequences are of strategic importance in modern biology. The authors make use of the resource to address the endocrine control of an ecologically and developmentally relevant life cycle transition, metamorphosis. As opposed to amphibian and flat fish where body plan changes, fish metamorphosis is anatomically more subtle and much less known, although it is clear that thyroid hormone (TH) signaling is a key player. The authors thus provide a repertoire of TH-relevant gene expression changes during development and across post-embryonic transitions and correlate developmental stages with changes of gene expression. Overall, this work represents a significant advance in the field.

Fish 'metamorphosis' is well known because it is not as spectacular as amphibians. This work clearly provides technical and theoretical resources to address in a more systematic manner the molecular changes occurring during development and post-embryonic transitions. Heterochrony is a major source of functional and life cycle diversity in fish, which blurs our anatomy-based understanding of fish biology, and has a direct impact on the protocols and rearing procedures used to produce live stocks. This work illustrates how, by using genomics coupled to simple experimental endocrinology, one directly addresses these challenges.

---

## [Author Response]

The following is the authors’ response to the original reviews.

**Responses to recommendations**

**Reviewer #1 (Recommendations For The Authors):**
Describe more precisely how gene expression graphs are built (tissues, reads counts). For example, how were read counts normalized? Were they from DESeq2 data, which only works by comparing two samples? If so, all samples should be independently compared to a reference and the normalized expression value of the reference will change from sample to sample... thus introducing a pure technical artifact.

We have added additional information about the normalisation method to the

Material and Methods section (Lines 597-598: “Lastly, expression levels shown in figures 2-5 are normalised gene counts produced by DESeq2.”) and figure legends

(lines 247, 286, 372, 404: “Gene expression data was generated from whole fish. Expression levels were derived from DESeq2 normalised gene counts.”) to address this recommendation.

DESeq2 provides a reference independent normalisation through a median of ratios method a good explanation can be found here:

https://hbctraining.github.io/DGE_workshop/lessons/02_DGE_count_normalization.html. The normalised expression values are independent of any reference, and therefore will not change from sample and sample as suggested in this comment. In contrast, the pairwise comparisons are done when analysing significantly differentially expressed genes between two treatments using a Wald test, which is done against a reference and generates log2 fold change information and p-values.; however, this is different to the normalisation we described above.

Provide bioinformatics workflows and, if possible, the set of parameters used, the computing resources, etc. Were some assembly finishing steps carried out (by long-range PCR?) and experimental validations (especially for allelespecific transcripts, by conventional RT-PCR based on diagnostic mutations)?

We have added additional information on the bioinformatics workflows where required, including parameters used (Lines 530, 536, 549-551, and 574-583.). No finishing steps other than HiC scaffolding were performed. No allele-specific analysis was done as part of this manuscript.

To further improve transparency, we have also uploaded all the scripts used for this study to https://github.com/R-Huerlimann/Malabar_grouper_genome and the gene models and functional annotation to https://figshare.com/projects/Malabar_grouper_Epinephelus_malabaricus_genome_ annotation/199909. This information has been added to the manuscript in lines 600601 and 609-611.

**Reviewer #3 (Recommendations For The Authors):**
General author response:All the recommendations of this reviewer are very relevant and would certainly provide a lot of information, but they are constituting a full project in themselves as they would imply establishing this grouper species as an experimental model in our lab. Currently we only have access to the larval and juvenile stages via a collaboration with the Okinawa Prefectural Sea Farming Center, which is an hour drive from our lab, and is limited to the grouper spawning season. If we want to do all what is suggested, we need to have a regular and easy access to the fishes. This would require establishing this model in our marine station, which is not possible due to space and time issues. These groupers grow to a very large size (1-2 m in length, and up to 150 kg in weight) and only mature into males after > 6 years.First and foremost, I would advise the authors to extend their TH and cortisol levels measurements to the entire developmental time considered in their analysis.

For the reasons stated above we could not perform these experiments. We must emphasize that the data regarding TH are available for a closely related species (e.g., Epinephelus coioides, de Jesus et al. 1998) and there is no reason to think that the situation will be drastically different in E. malabaricus. In addition, given that we have now studied several coral reef fish species in the same context (clownfish, surgeonfish, damselfish, gobies) we observed that the transcriptomic data are more robust, more sensitive, and more precise than hormone measurements.

Consider carrying out in situ hybridisation of TSH with putative CRH receptors to determine if thyrotrophin could be competent to respond to HPA axis signals.

We agree studying the interplay between corticoids and thyroid hormones at the neuroendocrine level would be desirable and we fully agree with the experiment suggested by the reviewer, but this is impossible in our current situation. We are not working with an establish animal model like zebrafish or *Xenopus*, but with a large, long-lived marine fish that reproduces in spawning aggregations and whose husbandry is notoriously difficult.

Consider conducting cortisol treatment experiments to functionally determine if indeed cortisol is involved in grouper metamorphosis.

We tried to do TH and cortisol treatments specifically on the early larval stages corresponding to the early TH peak to see how this would impact the development of the fin spines, but our trials were unsuccessful. The larvae at that stage are extremely fragile and even putting them into small volumes of treatment drugs induced massive mortalities. Again, this would mean establishing this grouper species as a model organism and would require a massive effort to improve larval rearing as discussed above. We feel that our data stands on its own in the meantime and adds valuable information to the existing literature by studying a rarely investigated species.

**Responses to comments**

**Reviewer #1 (Public Review):**
Weaknesses:The manuscript needs proper editing and is not complete. Some wordings lack precision and make it difficult to follow e.g. line 98 "we assembled a chromosome-scale genome of ..." should read instead "we assembled a chromsome-scla genome sequence of ...". Also, panel Figure 2E is missing.

We made the suggested change of adding “sequence” in lines 32 and 121. Concerning additional changes, we have carefully edited our manuscript and looked for any incomplete sections. Unfortunately, it is difficult to see what other issues are being raised here without any further information.

As for panel E of figure 2, it is not missing. The panel is located to the right, just below “Target Cells”.

The shortcomings of the manuscripts are not limited to the writing style, and important technical and technological information is missing or not clear enough, thereby preventing a proper evaluation of the resolution of the genomic resources provided:Several RNASeq libraries from different tissues have been built to help annotate the genome and identify transcribed regions. This is fine. But all along the manuscript, gene expression changes are summarized into a single panel where it is not clear at all which tissue this comes from (whole embryo or a specific tissue ?), or whether it is a cumulative expression level computed across several tissues (and how it was computed) etc. This is essential information needed for data interpretation.

No fertilised eggs or embryos have been sequenced. The individual tissues derived from juvenile fish were used for the genome annotation only, using ISOseq. The whole larval fish were used for the developmental analysis using RNAseq, as well as the genome annotation. We have added additional information in the figures and text that the results shown are from whole larvae, and added more detail to the material and methods section about which type of sample was analysed in which way.

Specifically, we have added “Lastly, expression levels shown in figures 2-5 are normalised gene counts produced by DESeq2.” to lines 597-598 in the Material and Methods section, “Gene expression data was generated from whole larvae.” to line 191, and “Gene expression data was generated from whole fish. Expression levels were derived from DESeq2 normalised gene counts.” to the figure legends in lines 247, 286, 372, 404. Additionally, we have added clarifications in lines 489, 497, 530, and 536.

The bioinformatic processing, especially of the assemble and annotation, is very poorly described. This is also a sensitive topic, as illustrated by the numerous "assemblathon" and "annotathon" initiatives to evaluate tools and workflows. Importantly, providing configuration files and in-depth description of workflows and parameter settings is highly recommended. This can be made available through data store services and documents even benefit from DOIs. This provides others with more information to evaluate the resolution of this work. No doubt that it is well done,but especially in the field of genome assembly and annotation, high resolution is VERY cost and time-intensive. Not surprisingly, most projects are conditioned by trade-offs between cost, time, and labor. The authors should provide others with the information needed to evaluate this.

We have added additional information on parameters used in the genome assembly, annotation and transcriptome analysis in lines 549-551, 577, 579, 580, and 582. Additionally, we have uploaded all scripts to github as outlined in the Code and Data Availability section (lines 599-614).

The genome assembly did not use a specific workflow (e.g., nextflow), but was done with a simple command and standard parameters in IPA. Scaffolding was carried out by Phase Genomics using their standardised proprietary workflow, of which a detailed description provided by Phase Genomics can be found in the supplementary material.

Quantifications of T3 and T4 levels look fairly low and not so convincing. The work would clearly benefit from a discussion about why the signal is so low and what are the current technological limitations of these quantifications.This would really help (general) readers.

The T3/T4 levels are consistent with other published work in fish. In the present manuscript for grouper we have a peak level of 1.2 ng/g (1,200 pg/g) of T4 and 0.06 ng/g (60 pg/g) of T3. This is a higher level of T4 and comparable level of T3 to what was found in convict tang (Holzer et al. 2017; Figure 2) with 30 pg/g of T4 and 100 pg/g of T3. Of course, there are also examples with higher levels, such as clownfish (Roux et al. 2023; Figure 1), with 10 ng/g (10,000 pg/g) of T4 and 2 ng/g (2,000 pg/g) of T3.

The differences could be due to different structure of fish tissues and therefore different hormone extraction efficiency, different hormone measurement protocols, different fish physiology, different fish size (e.g., the weighting of tiny grouper larvae is difficult and less precise than in convict tang). What is important is not the absolute level but the relative level, which shows the change within different larval stages of a species with identical extraction and measurement protocols. Which means our data is internally consistent and coherent with what the grouper literature says.

Holzer, Guillaume, et al. "Fish larval recruitment to reefs is a thyroid hormonemediated metamorphosis sensitive to the pesticide chlorpyrifos." Elife 6 (2017): e27595.

Roux, Natacha, et al. "The multi-level regulation of clownfish metamorphosis by thyroid hormones." Cell Reports 42.7 (2023).

Differential analysis highlights up to ~ 15,000 differentially expressed genes (DEG), out of a predicted 26k genes. This corresponds to more than half of all genes. ANOVA-based differential analysis relies on the simple fact that only a minority of genes are DEG. Having >50% DEG is well beyond the validity of the method. This should be addressed, or at least discussed.

The large number of differentially expressed genes is due to the fact that this is coming from a larval developmental transcriptome going from one day old larva to fully metamorphosed juveniles at around day 60.

While DESeq2 indeed works on an assumption that most genes are not differentially expressed, this affects normalization but not hypothesis testing (Wald-test, LRT tests or ANOVA). However, normalisation in DESeq2 is fairly robust to this assumption. According to the author of DESeq2, Micheal Love, DESeq2 is using the median ratio for normalisation, and as long as the number of up and down regulated genes is relatively even, DESeq2 will be able to handle the data. As part of our general quality control for this project we consulted the MA plots, which do not show any overrepresented up or down expression patterns. Additionally see Michael Love comment on comparing different tissues, which is also applicable here when comparing vastly different larval stages (https://support.bioconductor.org/p/63630/):

“For experiments where all genes increase in expression across conditions, the median ratio method will not be able to capture this difference, but this is typically not the case for a tissue comparison, as there are many "housekeeping" genes with relatively similar expression pattern across tissues.”

**Reviewer #3 (Public Review):**
Weaknesses:However, the authors make substantial considerations that are not proven by experimental or functional data. In fact, this is a descriptive study that does not provide any functional evidence to support the claims made.

We agree with the reviewer that our paper lacks functional experiments but despite that, the transcriptomic data clearly show the activation of TH and corticoid pathways during two distinct periods: an early activation between D1 and D10, and a second one between D32 and juvenile stage. These data are interesting as they call for further examination of (1) the existence of an early larval developmental step also involving TH and corticosteroids and (2) the possible interaction of corticoids and TH during metamorphosis. This is a question that is certainly not settled yet in teleost fishes and which is of great interest.

Especially (1) is of interest and importance, since this early activation (unique to our knowledge in any teleost fish studied so far) raises a lot of new questions and once again will certainly be scrutinised by other groups in the years to come, therefore ensuring a good citation impact of this study. We hope that the reviewer, while disagreeing with some our statements, will recognize that our study will be stimulating at that level and that this is what scientific studies should do.

We acknowledge the descriptive nature of the data and the lack of functional experiments in the Discussion in lines 443 to 445: “This may suggest that in some aspect, cortisol synthesis could work in concert with TH, as has been shown in several different contexts in amphibians, but functional experiments need to be conducted to confirm this hypothesis.” As stated above doing such functional experiment would require establishing the grouper as an experimental model in our husbandry, which currently is not possible due to the large size of the adult fish.

The consideration that cortisol is involved in metamorphosis in teleosts has never been shown, and the only example cited by the authors (REF 20) clearly states that cortisol alone does not induce flatfish metamorphosis. In that work, the authors clearly state that in vivo cortisol treatment had no synergistic effect with TH in inducing metamorphosis. Moreover, in Senegalensis, the sole pre-otic CRH neuron number decreases during metamorphosis, further arguing that, at least in flatfish, cortisol is not involved in flatfish metamorphosis (PMID: 25575457).

We will do our best to improve the clarity of the revised manuscript to avoid any misunderstanding about our claims. However, we would like to point out the semantic shift in the reviewer first sentence: Indeed “being involved” is not the same as “cortisol alone does not induce”. In ref 20 the authors explicitly wrote that “Cortisol further enhanced the effects of both T4 and T3, but was ineffective in the absence of thyroid hormones” and in our view this indeed corresponds to ”being involved in metamorphosis”.

We are not claiming that cortisol alone is involved in metamorphosis as the reviewer suggests, but simply that there is a possible involvement of cortisol together with TH in metamorphosis. We stand on this claim as we indeed observed an activation of corticoid pathway genes around D32, which is sufficient to say it is involved. We do agree that functional experiments will be needed to properly demonstrate the involvement of corticoids in grouper metamorphosis, but this was not possible in the current study as it would imply to set up a full grouper life cycle in lab conditions which is impossible for the scope of this manuscript.

We also mentioned in the discussion that the role of corticoids in fish larval development is still debated, and we agree that this remains a contentious issue. We have clarified the Discussion on this point (lines 375-376, lines 439-464).

We wrote that “There is contrasting evidence of communication between these two pathways during teleost fish larval development with some data suggesting a synergic and other an antagonistic relationship. In terms of synergy, an increase in cortisol level concomitantly with an increase in TH levels has been observed in flatfish [26], golden sea bream [64] and silver sea bream [65]. Cortisol was also shown to enhance in vitro the action of TH on fin ray resorption (phenomenon occurring during flatfish metamorphosis) in flounder [27]. It has also been shown that cortisol regulates local T3 bioavailability in the juvenile sole via regulation of deiodinase 2 in an organ-specific manner [66]. On the antagonistic side, it has been shown that experimentally induced hyperthyroidism in common carp decreases cortisol levels [67], whereas cortisol exposure decreases TH levels in European eel [68]. Given this scattered evidence, the existence of a crosstalk active during teleost larval development and metamorphosis has never been formally demonstrated. The results we obtained in grouper are clearly indicating that HPI axis is activated during both early development and metamorphosis and that cortisol synthesis is activated during early development. This may suggest that in some aspect, cortisol synthesis could work in concert with TH, as has been shown in several different contexts in amphibians [25], but functional experiments need to be conducted to confirm this hypothesis.” In the revised manuscript, we have also added the interesting case of the Senegal sole mentioned by the reviewer.

In the last revision, we had also added that our results “brought a first insight into the potential role of corticoids in the metamorphosis of E. malabaricus and call for functional experiments directly testing a possible synergy” meaning that we clearly acknowledge that we are only revealing a hypothesis that remains to be tested. We later follow up with a discussion about the most novel observation and focus of our study, the increase in THs and cortisol during early development, which was unexpected and very intriguing. Again, these results suggest that there might be a link between the two, as has been shown in amphibians. This is typically the kind of results that should encourage more investigations into other fish species. Indeed, this has been pointed out by other authors and in particular by Bob Denver (probably the foremost expert on this topic) in Crespi and Denver 2012: “Elevation in HPA/I axis activity has been described prior to Metamorphosis in amphibians and fish, birth in mammals (reviewed in Crespi & Denver 2005a; Wada 2008)”. B. Denver also adds that: “Experiments in which GCs were elevated prior to metamorphosis or prior to hatching or birth (e.g. Weiss, Johnston & Moore 2007) or inhibited by treatments with GC synthesis blockers (e.g. metyrapone) or receptor antagonists (e.g. RU486, Glennemeir & Denver 2002) demonstrate that GCs play a causal role in precipitating these life-history transitions (also reviewed in Crespi & Denver 2005a; Wada 2008).” We believe the reviewer will be convinced by these elements coming from a colleague unanimously respected in the field.

Furthermore, the authors need to recognise that the transcriptomic analysis is whole-body and that HPA axis genes are upregulated, which does not mean they are involved in regulating the HPT axis. The authors do not show that in thyrotrophs, any CRH receptor is expressed or in any other HPT axis-relevant cells and that changes in these genes correlate with changes in TSH expression. An in-situ hybridisation experiment showing co-expression on thyrotrophs of HPA genes and TSH could be a good start. However, the best scenario would be conducting cortisol treatment experiments to see if this hormone affects grouper metamorphosis.

We agree that functional experiments are needed to validate our hypothesis. As the early peaks of expression levels observed for many genes were very intriguing for us, we did carry out thyroid hormones and goitrogenic treatment on young grouper larvae to test their effect on the morphological changes. Unfortunately, such experiments, already tricky on metamorphosing larvae, are even more risky on such tiny individuals just after hatching and we encountered high mortality rates. We must add that because we cannot establish a full grouper life cycle under lab conditions, we have done these experiments in the context of a commercial husbandry system in Japan, which while excellent limits the scope of possible experiments. We were thus not able to provide functional validation of our hypothesis. Such experiments will be a full project in itself, requiring setting up a rearing system suitable for both larval survival and economical constraints related to drug treatments. We were further limited by the spawning times of the grouper in the operational aquaculture farm, which are limited to a short time during each year. So even if we strongly agree with the necessity of conducting such experiments, we think that this is not in the scope of the present paper, but something future research can explore.

High TSH and Tg levels usually parallel whole-body TH levels during teleost metamorphosis. However, in this study, high Tg expression levels are only achieved at the juvenile stage, whereas high TSH is achieved at D32, and at the juvenile stage, they are already at their lowest levels.

This is exactly our point. We observe two peaks in TSH expression, one at D3 and one at D32. The peak at D3 coincides with high thyroid hormone levels on the same day, and while we have not measured TH at D32, existing literature shows that there is a peak in TH during that time (e.g., de Jesus et al., 1998). Similarly, there is a small peak of Tg at D3. Our manuscript focused more on the upregulation of these genes at D3, which has not been reported before in the literature and raised the question of the role of TH so early in the larval development, outside of the metamorphosis period.

Regarding the respective levels of TSH and Tg, we first would like to add that their respective order of appearance before metamorphosis (TSH at D32, Tg after) is consistent with what we would expect. We agree however that the strong increase of Tg and TPO expression is later than expected. Therefore, we have added the following sentence in lines 212 to 216: “The respective order of appearance of TSH and Tg (TSH at D32, Tg after) is consistent with what we would expect but a bit later than expected given the morphologicl transformation. It would be interesting to revisit this in a future series of experiments, with tighter temporal sampling to study how gene expression and morphological transformation aligned.“.

It is very difficult to conclude anything with the TH and cortisol levels measurements. The authors only measured up until D10, whereas they argue that metamorphosis occurs at D32. In this way, these measurements could be more helpful if they focus on the correct developmental time. The data is irrelevant to their hypothesis.

We respectfully disagree with the reviewer, considering that (1) TH levels have already been investigated in groupers coinciding with pigmentation changes and fin rays resorption (Figure 4 in de Jesus et al, 1998), (2) there is also evidence in numerous fish species that TH level increase is concomitant with increase of TH related genes, and (3) we observed in our data an increase in the expression of TH related genes as well as pigmentation changes and fin rays resorption. Based on our experience in fish metamorphosis and the literature we can say confidently that those observations indicate that metamorphosis is occurring between D32 and the juvenile stage. This clearly shows that our inference is correct. Additionally, we would like to reemphasize that from our experience in several fish species transcriptomic data are more robust and precise than hormone measurements.

However, as we were surprised by the activation of TH and corticoid pathway genes very early in the larval development (at D3), which is clearly outside of the metamorphosis period, we decided to measure TH and cortisol levels during this period of time to determine if whether or not there this surprising early activation was indeed corresponding to an increase in both TH and cortisol. As such observation has never been made in other teleost species (to our knowledge), and as we were wondering if gene activation was accompanied by hormonal increase, the measurements we did for TH and cortisol between D1 and D10 are relevant. In order to clarify our message further, we have changed some of the mentions of

“metamorphosis” to “larval development” throughout the manuscript and added other improvements to avoid any confusion between the two periods we are studying: early larval development (between D1 and D10) and metamorphosis (between D32 and juvenile stage).

Moreover, as stated in the previous review, a classical sign of teleost metamorphosis is the upregulation of TSHb and Tg, which does not occur at D32 therefore, it is very hard for me to accept that this is the metamorphic stage. With the lack of TH measurements, I cannot agree with the authors. I think this has to be toned down and made clear in the manuscript that D32 might be a putative metamorphic climax but that several aspects of biology work against it. Moreover, in D10, the authors show the highest cortisol level and lowest T4 and T3 levels. These observations are irreconcilable, with cortisol enhancing or participating in TH-driven metamorphosis.

We thank the reviewer for this comment, but we think that there might be a misunderstanding here.

(1) We clearly observed an increase of TSHb (that occurs between D18 and juvenile stage) and an increase of tg from D32 which coincide with the activation of other genes involved in TH pathway (dio2, dio3, and also a strong increase of TRb). All this and put in the context of what we know from previous grouper studies, clearly supports our conclusion that TH-regulated metamorphosis is starting at around D32 in grouper. We also observed morphological changes such as fin rays resorption and pigmentation changes between D32 and juvenile stage. Such morphological changes have already been associated as corresponding to metamorphosis in groupers (De Jesus et al 1998) as they occur during TH level increase, and they also happen to be under the control of TH in grouper (De Jesus et al 1998). Based on this study but also on studies (conducted on many other teleost species) showing that the increase of TH levels is always associated with an activation of TH pathway genes and morphological and pigmentation changes we concluded that metamorphosis of E. malabaricus occurs between D32 and juvenile stage. We have improved the clarity of the manuscript in several places to make sure that our conclusion is based on our transcriptomic and morphological data plus the available literature.

(2) We clearly observed another activation of TH related gene earlier in the development between D1 and D10, with a surge of trhrs, tg and tpo at D3. As this activation was very unexpected for us, we decided to focus the analysis of TH levels between D1 and D10 and very interestingly we observed high level of T4 at D3 indicating that THs are instrumental very precociously in the larval development of the malabar grouper which has never been shown before. We declared lines 224-225 that our “data reinforce the existence of two distinct periods of TH signalling activity, one early on at D3 and one late corresponding to classic metamorphosis at D32”. However, we agree that we could have been clearer and clearly explained that this early activation was very intriguing for us and that we wanted to investigate hormonal levels around that period. However, we never claimed anywhere in the manuscript

that this early developmental period corresponds to metamorphosis. Something else is occurring and both TH and cortisol seem to be involved but further experiments need to be conducted to understand their role and their possible interaction. We have added corresponding statements in the abstract (lines 39-43) and discussion (lines 447 to 449).

(3) Finally, regarding the comment about cortisol enhancing or participating in TH driven metamorphosis, our data clearly showed an activation of the corticoid pathway genes around metamorphosis (between D32 and juvenile stage) suggesting a potential implication of corticoids in metamorphosis, but we agree with the reviewer that further experiment are needed to test that. We never claimed that cortisol was enhancing or participating in metamorphosis, on the contrary we are “suggesting a possible interaction between TH and corticoid pathway during metamorphosis”. And we also say that our “results brought a first insight into the potential role of corticoids in the metamorphosis of E. malabaricus and call for functional experiments directly testing a possible synergy.” Nonetheless, we agree that some parts of our manuscript can be confusing in regards of cortisol synthesis during metamorphosis as we did not measure cortisol levels between D32 and juvenile stage. We have therefore made changes throughout the Introduction and Discussion to make this clearer.

Given this, the authors should quantify whole-body TH levels throughout the entire developmental window considered to determine where the peak is observed and how it correlates with the other hormonal genes/systems in the analysis.

We did not measure TH levels at later stages as it has already been measured during Epinephelus coioides metamorphosis and the morphological changes observed in this species around the TH peak corresponds to what we observed in Epinephelus malabaricus around the peak of expression of TH pathway genes (see De Jesus et al., 1998 General and Comparative Endocrinology, 112:10-16). The main focus of this manuscript is the novel observation of the existence of an early activation period observed at D3, and for which we needed TH levels to determine if they were involved in another early developmental process (not related to metamorphosis). Our hypothesis is that this early activation might be related to the growth of fin rays necessary to enhance floatability during the oceanic larval dispersal. As we may have arrived at the explanation of this hypothesis too rapidly without setting up the context well enough, we have made changes to the introduction and discussion.

Even though this is a solid technical paper and the data obtained is excellent, the conclusions drawn by the authors are not supported by their data, and at least hormonal levels should be present in parallel to the transcriptomic data. Furthermore, toning down some affirmations or even considering the different hypotheses available that are different from the ones suggested would be very positive.

We thank the reviewer for acknowledging the solidity of the method of our paper and the quality of the results. We agree that there were several parts where our message was unclear. We have addressed these points in the revised version of the manuscript to make sure there is no more confusion between the two distinct periods we studied in this paper (early larval development and metamorphosis). We also made sure that our claims about TH/corticoids interaction during both periods remain hypothetical as we cannot yet, despite trials, sustain them with functional experiment.